# Mesoscopic description of hippocampal replay and metastability in spiking neural networks with short-term plasticity

**Bastian Pietras**[1,2,3], **Valentin Schmutz**[4], **Tilo Schwalger**[1,2]*

**1** Institute for Mathematics, Technische Universität Berlin, Berlin, Germany, **2** Bernstein Center for Computational Neuroscience, Berlin, Germany, **3** Department of Information and Communication Technologies, Universitat Pompeu Fabra, Barcelona, Spain, **4** Brain Mind Institute, School of Computer and Communication Sciences and School of Life Sciences, École Polytechnique Fédérale de Lausanne (EPFL), Lausanne, Switzerland

* schwalger@math.tu-berlin.de

**Data Availability Statement:** All relevant data are within the manuscript and its Supporting information files. The code used to numerically solve the equations derived in the manuscript is

## Abstract

Bottom-up models of functionally relevant patterns of neural activity provide an explicit link between neuronal dynamics and computation. A prime example of functional activity patterns are propagating bursts of place-cell activities called hippocampal replay, which is critical for memory consolidation. The sudden and repeated occurrences of these burst states during ongoing neural activity suggest metastable neural circuit dynamics. As metastability has been attributed to noise and/or slow fatigue mechanisms, we propose a concise mesoscopic model which accounts for both. Crucially, our model is bottom-up: it is analytically derived from the dynamics of finite-size networks of Linear-Nonlinear Poisson neurons with short-term synaptic depression. As such, noise is explicitly linked to stochastic spiking and network size, and fatigue is explicitly linked to synaptic dynamics. To derive the mesoscopic model, we first consider a homogeneous spiking neural network and follow the temporal coarse-graining approach of Gillespie to obtain a "chemical Langevin equation", which can be naturally interpreted as a stochastic neural mass model. The Langevin equation is computationally inexpensive to simulate and enables a thorough study of metastable dynamics in classical setups (population spikes and Up-Down-states dynamics) by means of phase-plane analysis. An extension of the Langevin equation for small network sizes is also presented. The stochastic neural mass model constitutes the basic component of our mesoscopic model for replay. We show that the mesoscopic model faithfully captures the statistical structure of individual replayed trajectories in microscopic simulations and in previously reported experimental data. Moreover, compared to the deterministic Romani-Tsodyks model of place-cell dynamics, it exhibits a higher level of variability regarding order, direction and timing of replayed trajectories, which seems biologically more plausible and could be functionally desirable. This variability is the product of a new dynamical regime where metastability emerges from a complex interplay between finite-size fluctuations and local fatigue.

available at https://github.com/BastianPietras/meso_stp.

**Funding:** BP received funding from the European Union's Horizon 2020 research and innovation programme under the Marie Sklodowska-Curie grant agreement no. 101032806 (https://marie-sklodowska-curie-actions.ec.europa.eu/). VS received funding by the Swiss National Science Foundation (grant no. 200020 184615, www.snf.ch). The funders had no role in study design, data collection and analysis, decision to publish, or preparation of the manuscript.

**Competing interests:** The authors have declared that no competing interests exist.

## Author summary

Cortical and hippocampal areas of rodents and monkeys often exhibit neural activities that are best described by sequences of re-occurring firing-rate patterns, so-called metastable states. Metastable neural population dynamics has been implicated in important sensory and cognitive functions such as neural coding, attention, expectation and decision-making. An intriguing example is hippocampal replay, i.e. short activity waves across place cells during sleep or rest which represent previous animal trajectories and are thought to be critical for memory consolidation. However, a mechanistic understanding of metastable dynamics in terms of neural circuit parameters such as network size and synaptic properties is largely missing. We derive a simple stochastic population model at the mesoscopic scale from an underlying biological neural network with dynamic synapses at the microscopic scale. This "bottom-up" derivation provides a unique link between emergent population dynamics and neural circuit parameters, thus enabling a systematic analysis of how metastability depends on neuron numbers as well as neuronal and synaptic parameters. Using the mesoscopic model, we discover a novel dynamical regime, where replay events are triggered by fluctuations in finite-size neural networks. This fluctuation-driven regime predicts a high level of variability in the occurrence of replay events that could be tested experimentally.

## Introduction

Metastable dynamics of neural populations is an important concept in computational neuroscience with increasing experimental evidence [1, 2]. It is loosely defined as a sequence of recurring, discrete "states" of population activity that last much longer than the rapid, jump-like transitions between states (typically hundreds of milliseconds to several seconds). Sequences of metastable states have been frequently observed in cortical and hippocampal areas during task engagement as well as during spontaneous, ongoing activity and have been linked to various sensory and cognitive functions [3, 4]. These functions include the encoding of sensory stimuli [5, 6] and internal representations of expectation [7] and attention [8]. In these studies, the statistical properties of metastable neural activity can often be explained by hidden Markov models with a few latent states [6, 8]. However, more complex spatio-temporal activity patterns such as sequences of burst activity across hippocampal place cells during periods of both exploration and immobility ("replay of trajectories") of an animal can also be regarded as metastable activity. In in-silico studies, metastable dynamics also emerges in networks of excitatory and inhibitory spiking neurons. This is the case for finite-size networks with clustered connectivity [6, 9], spatially-structured networks with slow fatigue processes for hippocampal replay [10] and even for unstructured random connectivity in the inhibition-dominated regime [11, 12]. Network models exhibiting metastable dynamics have also been used to explain the stimulus-dependence of cortical variability [9].

The mechanisms of metastable dynamics are often explained using heuristic population, or firing-rate, models. These mechanisms can be roughly divided into two types: one in which transitions between metastable states are induced by fluctuations and another one in which transitions are induced by the deterministic part of the dynamics. In the first case, noise is essential for metastability because the noiseless dynamics would not exhibit spontaneous transitions. In contrast, in the second case, transitions also occur in the noiseless dynamics, while noise can still be useful to model variability of state durations. An important instance for the first type are multi-attractor models in the presence of noise, such as noisy bistable models for

perceptual rivalry [13–15] and alternating Up-Down states [16–18]. Transitions correspond to noise-induced escapes from the basins of attraction. A popular instance of the second type are transiently stable states governed by a slow fatigue variable such as adaptation or synaptic depression. In these fast-slow systems, rapid transitions occur when quasi-stationary states of the fast subsystem (e.g. population activity) destabilize or vanish as the slow subsystem (fatigue process) evolves on a longer time scale. A prototypical example are relaxation oscillations, i.e. a (noisy) limit-cycle with a strong time-scale separation, used e.g. to model regular alternations between Up and Down states in spontaneous cortical activity [19, 20]. A complex example of fatigue-induced metastability is the Romani-Tsodyks ring model for nonlocal events in place cells resembling the hippocampal "replay" dynamics [21]. This deterministic ring model resides in a traveling-wave state ("non-local events") or in a quiescent state depending on the spatial profile of a slow synaptic depression variable, which leads to a complex spatio-temporal activity pattern. We mention that there are also other mechanisms of metastability including noisy excitable dynamics [20] and deterministic motions between saddle points [3], also referred to as heteroclinic cycles or winnerless competition [22]. Looking at empirical data, however, it can be hard to distinguish different mechanisms, especially at high noise levels.

There has been much effort to infer the mechanism underlying metastable dynamics by studying the consistency of experimental data with heuristic population models. For example, in the case of cortical and hippocampal Up and Down states [18–20] and for perceptual bistability [13–15], it has been suggested that population models, where noise-induced transitions are modulated by a slow fatigue variable, are most consistent with the data. An important question that has received relatively little attention is whether such conclusions are also consistent with the underlying circuit properties at the microscopic scale, modeled as networks of spiking neurons with biologically realistic neuronal, synaptic and network properties. Unfortunately, a clear link between the employed population models and microscopic circuit models is largely missing, and it thus remains unclear how the mechanisms of metastability depend on physiological parameters. While neuronal and synaptic properties can be accounted for by mean-field models of integrate-and-fire networks [6, 23], the dependence of metastable dynamics on the number $N$ of neurons in the network is poorly understood. This latter aspect is particularly crucial in the context of metastability because fluctuations due to a finite number of neurons have been found to be essential for fluctuation-induced metastability by several detailed simulation studies [6, 9] as well as theoretical considerations [24]. The description of these internally generated fluctuations requires population models at the *mesoscopic* scale, where the finite network size is explicitly taken into account [25–27]. Previous models for determining the mechanisms of metastability cannot describe this dependence: In heuristic population models, fluctuations were introduced *ad hoc* by adding a phenomenological noise term without a link to the network size $N$. In the case of mean-field models, fluctuations are usually not described at all because they vanish in the mean-field limit of infinitely many neurons, $N \to \infty$. We will refer to such deterministic mean-field models as *macroscopic* models.

In this contribution, we develop a theoretical framework for mesoscopic population dynamics with slow fatigue that can describe metastable dynamics and links to an underlying microscopic description. To this end, we use a bottom-up approach starting from a finite-size network of linear-nonlinear Poisson (LNP) spiking neurons connected via dynamic synapses undergoing short-term plasticity (STP). From this microscopic model we derive stochastic differential equations for a few mesoscopic variables describing the coarse-grained population dynamics. We focus on STP in the form of short-term synaptic depression as a slow fatigue mechanism because it is a ubiquitous feature of neural networks in the brain [28–32] and has been implicated in important functions such as temporal filtering [33, 34], multistability [35] and working memory [36]. Mean-field models of STP [23] have recently gained renewed

attention [37, 38] in the context of the Montbrió-Pazó-Roxin theory for quadratic integrate-and-fire neurons [39, 40]. These macroscopic models are exact in the case of static synapses, or when STP is introduced at the population level, but not when dynamic synapses undergo STP individually. Moreover, the resulting firing rate equations are deterministic—as they hold in the limit of infinitely many neurons independently of individual neurons being subject to Gaussian [41] or Cauchy noise [42, 43]—and thus cannot explain fluctuation-induced transitions between metastable states in finite-size networks. Recently, we have developed a mesoscopic bottom-up model for finite-size networks with STP and have demonstrated that the mesoscopic model accurately reproduces the metastable Up-and-Down-states dynamics of the microscopic model [44]. The mathematical structure of that model has the intricate form of a state-dependent doubly-stochastic point-process driving a system of stochastic differential equations. As such, it is difficult to analyze and it lacks a straightforward, efficient simulation algorithm. However, the mesoscopic theory of [44] can be used as a starting point to derive a temporally coarse-grained stochastic dynamics in the form of a simple jump-diffusion process. For the case of synaptic depression and large network size, we also present a short direct derivation of the diffusion limit that yields a mesoscopic model in the form a simple diffusion process.

As we shall show below, our bottom-up modeling framework for mesoscopic population dynamics permits a re-evaluation of existing heuristic models for metastability in terms of an underlying microscopic network model. As a first example, we consider a single population of excitatory neurons with synaptic depression that generates population spikes and can transition between Up and Down states. The corresponding mesoscopic population model is similar to the model by Holcman, Tsodyks and co-workers [16, 45, 46], which successfully reproduced experimental observations [46]. The important difference is that in our mesoscopic model all parameters are fixed by the microscopic parameters. Thanks to the low-dimensional character of the reduced mesoscopic system, we can apply phase-plane analysis to study the emergence of multiple stable states that soon become metastable when decreasing the network size. As a second, more complex example, we revisit the Romani-Tsodyks (RT) model for hippocampal replay activity in circular networks of place cells with synaptic depression [21]. We propose a spiking-neural-network implementation of the original firing-rate model. The corresponding mesoscopic population model with finite-size noise enables us to shed new light on the mechanisms underlying hippocampal replay in place cells of area CA3 in the hippocampus. In the deterministic (and heuristic) RT model, irregular switching between metastable traveling waves of sequential neural activity and a quiescent state is solely controlled by local synaptic depression as a slow fatigue mechanism [21]. Yet, it is unclear whether such metastable replay dynamics also occurs in finite-size networks of spiking neurons and whether, in this case, replay sequences are fatigue-induced (like in the RT model) or may also be driven by finite-size fluctuations. With our model that is reduced from a microscopic network of a finite number of neurons, we can interpolate between a fatigue-induced regime and a novel regime of fluctuation-induced hippocampal replay. We show that these two regimes lead to very distinct statistical predictions, which can be tested experimentally.

In the present paper, the Results are organized in three main parts. In the first part, we present the mesoscopic bottom-up model in two variants, a diffusion and a jump-diffusion model. In the second part, we use a single population model to demonstrate the performance of the two variants with respect to network size and dynamical regime. In the third part, we turn to the more complex scenario of metastable nonlocal replay events in hippocampal place cells. We compare a novel dynamical regime of fluctuation-induced replay with the deterministic replay dynamics of [21]. In the Discussion, we explicate biological limitations of the model and possible extensions to address these limitations. We also discuss potential advantages of

the novel fluctuation-induced replay dynamics. Finally, in the Methods section, we provide the derivation of the mesoscopic model as well as the details on numerical simulations and statistical procedures.

## Results

### Mesoscopic description of microscopic network dynamics

We study the dynamics of a network of $N$ spiking neurons that, on the microscopic level, are modeled as linear-nonlinear-Poisson (LNP) neurons [47–49] with dynamic synapses [44, 50]. The network of LNP neurons without dynamic synapses is also referred to as a multivariate nonlinear Hawkes process [51, 52] in the mathematical neuroscience literature. The LNP model consists of a cascade of three steps: a temporal linear filter acting on the synaptic input, a static nonlinearity, which accounts for nonlinearities of the output firing rate, and a stochastic spike generation mechanism. While the simplicity of the LNP model enables us to derive mesoscopic dynamics with spiking noise and dynamic synapses, we note that the model does not capture spike correlations due to refractoriness and other spike-history effects [47]. The synaptic dynamics is given by the Tsodyks-Markram model of short-term plasticity (STP) [23]. For simplicity, we focus here, in the Results part, on the special case where the synaptic dynamics corresponds to pure depression [53] and the linear filter of the LNP model corresponds to a leaky integration of the input [54]. The general theory for the full Tsodyks-Markram model with depression and facilitation as well as the straightforward extension to general linear filters, enabling biologically more realistic neuronal dynamics [54], is provided in the Methods part.

Specifically, each LNP neuron $i = 1, \ldots, N$ is characterized by an input potential $h_i(t)$, which can be loosely interpreted as its membrane potential at time $t$. Given the current value of $h_i(t)$, neuron $i$ emits a spike independently with state-dependent hazard rate $f(h_i(t))$, also known as conditional intensity [55]. Here, $f$ is a non-negative, increasing function that represents the static nonlinearity of the LNP model (see Eq (6) below for a concrete choice of $f$). Thus, the conditional probability that neuron $i$ fires a spike in an infinitesimally small time interval $[t, t + dt)$ given the input potential $h_i(t)$ is

$$\text{Prob}\{\text{spike of neuron } i \text{ in } [t, t + dt)|h_i(t^-)\} = f(h_i(t^-))dt, \tag{1a}$$

where $t^-$ denotes the time just before $t$. The corresponding spike train is the sum of Dirac delta pulses

$$s_i(t) = \sum_k \delta(t - t_k^i), \tag{1b}$$

where $\{t_k^i\}_k$ are the spike times of neuron $i$. Mathematically, these spike times are a realization of a stochastic point process with conditional intensity $f(h_i(t^-))$. In numerical simulations of networks of $N$ LNP neurons, in each small time step $[t, t + \Delta t)$, we draw $N$ independent Bernoulli random variables $z_{t,1}, \ldots, z_{t,N} \in \{0, 1\}$ with success probabilities $f(h_1(t^-))\Delta t, \ldots, f(h_N(t^-))\Delta t$, respectively. If $z_{t,i} = 1$, then neuron $i$ emits a spike at time $t$.

We further consider fully-connected networks with homogeneous connectivity and depressive synapses: each time neuron $i$ emits a spike, all the input potentials in the network make a jump proportional to $\frac{1}{N}x_i(t^-)$, where $x_i(t^-)$ is the amount of synaptic resources available at the outgoing synapses of neuron $i$ just before its spike. In contrast to standard leaky integrate-and-fire models [49], the potential $h_i(t)$ is not reset to a fixed value after each spike of neuron $i$. Together with Eqs (1a) and (1b), the microscopic network model with LNP neurons and

synaptic short-term depression (STD), in short LNP-STD model, reads

$$\frac{dh_i}{dt} = \frac{\mu(t) - h_i}{\tau} + \frac{JU_0}{N}\sum_{j=1}^{N}x_j(t^-)s_j(t), \tag{1c}$$

$$\frac{dx_i}{dt} = \frac{1 - x_i}{\tau_D} - U_0 x_i(t^-)s_i(t). \tag{1d}$$

Here, $\mu(t)$ represents a common external current mimicking, e.g., feedforward input from other areas, and $\tau$ can be interpreted as the membrane time constant. The synaptic parameters are given by the overall synaptic weight factor $J$, the relative depletion of neurotransmitter by a single transmitted spike $U_0$ and the time scale of synaptic depression $\tau_D$. Note that $h_i(t) = h_j(t)$ for all $t \geq 0$ and for all $i$ and $j$ if at time 0, all the $h_i$ share the same initial condition.

From a mathematical point of view, the microscopic model (1) can be rigorously interpreted as a system of Poisson-noise-driven stochastic differential equations with solutions taken in the sense of Itô.

**Diffusion model of the mesoscopic dynamics (Gaussian noise).** Our goal is to derive a mean-field model for the microscopic dynamics, Eq (1), that accounts both for the finite number of neurons as well as for the dynamic synapses undergoing Tsodyks-Markram STP, see Fig 1. The mean-field description will be based on the dynamics of the following mesoscopic variables defined as the empirical averages

$$h(t) := \frac{1}{N}\sum_{i=1}^{N}h_i(t), \quad x(t) := \frac{1}{N}\sum_{i=1}^{N}x_i(t), \quad Q(t) := \frac{1}{N}\sum_{i=1}^{N}x_i^2(t). \tag{2}$$

The desired dynamics of $h(t)$, $x(t)$ and $Q(t)$ are supposed to no longer depend on (the index $i$ of) individual neurons, so we will approximate terms such as, e.g., the sum $\frac{1}{N}\sum_{i=1}^{N}x_i(t^-)s_i(t)$, by a diffusion term which only involves the mesoscopic variables. To this end, we follow the temporal coarse-graining approach by Gillespie [56] for the derivation of a "chemical Langevin equation", see the Methods section for a detailed derivation. In brief, we first use a *macroscopically infinitesimal* time step $\Delta t$ [56] and approximate the coarse-grained sum

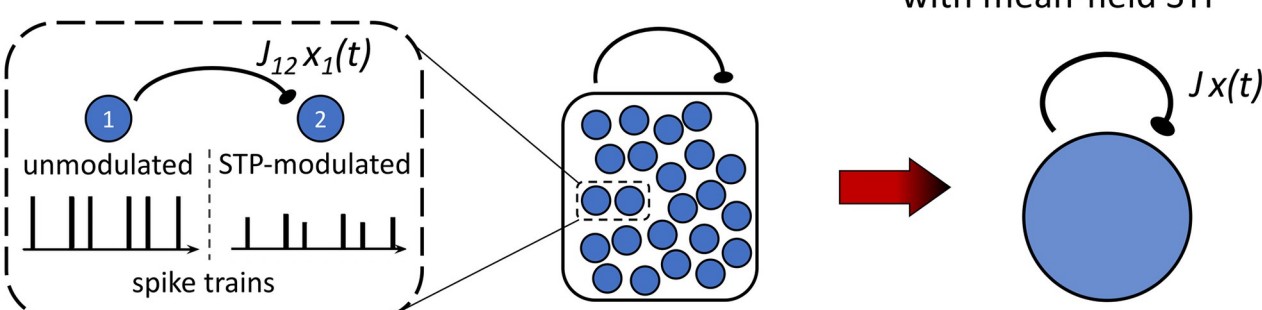

**A** Network with microscopic STP

**B** Mesoscopic population with mean-field STP

**Fig 1. From microscopic to mesoscopic population dynamics.** (A) Network with microscopic short-term plasticity. Dashed region shows a zoom into a pair of interconnected neurons: presynaptic neuron 1 sends out an unmodulated spike train to postsynaptic neuron 2 that receives the spike-train modulated by short-term depression. (B) Mesoscopic mean-field model with one effective synapse undergoing short-term depression.

$\int_t^{t+\Delta t} \frac{1}{N} \sum_{i=1}^N x_i(t^-) s_i(t)\, dt$ by a Gaussian random variable with variance proportional to $Q(t^-)$. In a second step, we derive the dynamics of $Q(t)$, discarding the fluctuations whose effect on $h(t)$ and $x(t)$ is of order $N^{-3/2}$. The resulting mesoscopic mean-field dynamics is given by the *diffusion model*

$$\frac{dh}{dt} = \frac{\mu(t) - h}{\tau} + JU_0 x f(h) + JU_0 \sqrt{\frac{Qf(h)}{N}} \xi(t), \tag{3a}$$

$$\frac{dx}{dt} = \frac{1 - x}{\tau_D} - U_0 x f(h) - U_0 \sqrt{\frac{Qf(h)}{N}} \xi(t), \tag{3b}$$

$$\frac{dQ}{dt} = 2\frac{x - Q}{\tau_D} - U_0(2 - U_0)Qf(h), \tag{3c}$$

where $\xi(t)$ is a Gaussian white noise with auto-correlation function $\langle \xi(t)\xi(s) \rangle = \delta(t - s)$. Although it is possible to deduce Eq (3) from the detailed doubly-stochastic mesoscopic dynamics derived in [44] (see Methods), the derivation summarized above and presented in Methods Section "Diffusion approximation for the mesoscopic dynamics with short-term depression" is much simpler as it relies on a direct application of the diffusion approximation (avoiding the detour via the model presented in [44]).

The stochastic differential equation (SDE) (3), as all the SDEs in this work, is integrated in the sense of Itô. Numerically, this means that for each integration time step, the factor $\sqrt{Qf(h)/N}$ multiplying the noise is taken at the time *just before* the random increment.

**Jump-diffusion model of the mesoscopic dynamics (Hybrid noise).**   In large networks, it is plausible to assume that the spike input through a large number of recurrent connections can be approximated by a Gaussian process, and the diffusion model Eq (3) is valid for sufficiently large $N$. In smaller networks, by contrast, we may no longer rely on the diffusion approximation since we need to take into account the shot noise character of the spike input. To this end, we start from the mesoscopic model of [44] and derive a mesoscopic jump-diffusion model with facilitation and depression (see Methods). In this model, the noise takes on a hybrid form combining Poisson shot noise and Gaussian white noise. In the special case of short-term synaptic depression only, the resulting *jump-diffusion model* of the mesoscopic dynamics reads

$$\frac{dh}{dt} = \frac{\mu(t) - h}{\tau} + JU_0 \left[ x(t^-)A(t) + \sqrt{\frac{yf(h)}{N}} \xi_x(t) \right], \tag{4a}$$

$$\frac{dx}{dt} = \frac{1 - x}{\tau_D} - U_0 \left[ x(t^-)A(t) + \sqrt{\frac{yf(h)}{N}} \xi_x(t) \right], \tag{4b}$$

$$\frac{dy}{dt} = -\left[ \frac{2}{\tau_D} + U_0(2 - U_0)f(h) \right] y + U_0^2 x^2 f(h), \tag{4c}$$

where we introduced the new variable $y(t) := Q(t) - x(t)^2$, which can be interpreted as the population variance of the variables $\{x_i(t)\}$. Furthermore, $\xi_x(t)$ is a Gaussian white noise with auto-

correlation function $\langle \xi_x(t)\xi_x(s) \rangle = \delta(t-s)$ and

$$A(t) = \frac{1}{N}\frac{dn(t)}{dt} = \frac{1}{N}\sum_k \delta(t-t_k), \qquad dn(t) \sim \text{Pois}\left[Nf(h(t^-))dt\right], \qquad (4d)$$

is a shot noise. The shot noise $A(t)$ is defined by the counting process $n(t)$ with jump times $t_k$ that occur with conditional intensity $Nf(h(t^-))$. This definition means that the probability for a jump in a sufficiently small time step $\Delta t$ is $Nf(h(t^-))\Delta t$ and the increments $\Delta n(t) = n(t+\Delta t) - n(t)$ of the counting process are conditionally independent given the values $h(t^-)$ just before time $t$. The increment of the counting process $dn(t)$ represents the total number of spikes generated by all neurons in the small time interval $[t, t+dt)$, and $A(t)$ is therefore the empirical population activity. The presence of two different sources of noise in Eq (4) can be interpreted as the effect of two components that make up the synaptic input $N^{-1}\sum_i x_i(t^-)s_i(t)$ on the mesoscopic scale: First, a term $(N^{-1}\sum_i x_i(t^-)) \cdot (N^{-1}\sum_i s_i(t)) = x(t^-)A(t)$ that arises if the variability of the weighting factors $x_i$ across synapses is neglected. This term represents the common spiking noise caused by shared spike inputs. Second, a correction term that accounts for the variability of $x_i$, approximated by a Gaussian distribution with variance $y(t)$ as shown previously [44].

Mathematically, the mesoscopic model Eq (4) is a jump-diffusion process because the shot noise leads to small jumps of order $1/N$ in addition to the diffusive dynamics caused by the Gaussian white noise. The jumps, however, occur at a high rate $Nf(h(t))$ so that in simulations with a coarse-grained time step $\Delta t$, unitary jumps will not be resolved. Instead, the increment of the spike count $\Delta n(t) = n(t+\Delta t) - n(t)$ can be drawn from a Poisson distribution with mean $Nf(h(t))\Delta t$ provided a sufficiently small simulation time step $\Delta t \ll 1/f(h)$, $\tau$, $\tau_D$. The population activity is then obtained from the increments as $A(t) = \Delta n(t)/(N\Delta t)$.

We expect that the jump-diffusion model Eq (4) remains valid for small network sizes, for which the diffusion model Eq (3) ceases to provide an accurate description of the microscopic network dynamics Eq (1). For large $N$, the jump-diffusion model Eq (4) converges to the diffusion model Eq (3), see also Methods "Reduction to a pure diffusion process".

**Mean-field model of the macroscopic dynamics (no noise).**   When the number of neurons becomes infinitely large, i.e. in the thermodynamic limit $N \to \infty$, fluctuations disappear on the population level and Eqs (3) or (4) becomes a skew system as the variable $Q$ no longer influences the dynamics of $h$ and $x$. The *macroscopic model* then reads

$$\frac{dh}{dt} = \frac{\mu(t) - h}{\tau} + JU_0 x f(h), \qquad (5a)$$

$$\frac{dx}{dt} = \frac{1-x}{\tau_D} - U_0 x f(h), \qquad (5b)$$

where we identified $A(t) = f(h(t))$ by taking the limit $N \to \infty$ in Eq (4d). Thus, the transfer function that relates the input potential $h(t)$ to the neuronal population activity $A(t)$ in the macroscopic model is exactly given by the static nonlinearity $f$ of the LNP neuron model, Eq (1). Put differently, the nonlinear function $f$ in the macroscopic model has a precise microscopic interpretation as the static nonlinearity of the microscopic model. While the two-dimensional macroscopic model, Eq (5), permits a phase-plane analysis of the underlying deterministic dynamics, it cannot describe fluctuation-induced transitions between metastable states.

## Microscopic vs. mesoscopic dynamics of a single population exhibiting metastability

The mesoscopic descriptions Eqs (3) and (4) of the full network of $N$ interacting spiking neurons with short-term depression (STD) effectively reduce the high-dimensional microscopic dynamics Eq (1) to a system of three stochastic differential equations in $h$, $x$, and $Q$. In the limit $N \to \infty$, finite-size fluctuations in the mesoscopic dynamics vanish and the variable $Q$ becomes superfluous. The two-dimensional macroscopic dynamics Eq (5) readily allows for a comprehensive phase-plane analysis, see, e.g., [16, 46], which reveals the deterministic backbone of, and can therefore yield theoretical insights about, the full network dynamics.

To demonstrate the high accuracy of our mesoscopic description and also its usefulness for studying the effect of finite-size fluctuations on metastable dynamics, we will focus in this Section on two traditional examples of metastability in a single excitatory population ($J > 0$) of LNP-STD neurons: populations spikes and spontaneous transitions between Up and Down states. For reasons of comparability with previous models [16, 21], we assume in this paper that the static nonlinearity of the microscopic LNP model (1) has the form

$$f(h) = ra \ln\{1 + \exp[(h - h_0)/a]\} \tag{6}$$

with slope parameter $r$, smoothness $a$, and threshold $h_0$. The function $f(h)$ has an exponential sub-threshold tail and a linear supra-threshold part [21, 57]. In the limit $a \to 0$, $f(h) = r[h - h_0]^+$ becomes a threshold linear function with slope $r$ and threshold $h_0$ [16]. The larger $a > 0$, the smoother the transition at the threshold.

**Population spikes in an excitatory population.**    As a first example, we study the emergence of spontaneous bursts of synchronized activity due to finite-size fluctuations and short-term depression (Fig 2). To this end, we tune the parameters of our model such that the macroscopic dynamics for $N \to \infty$ exhibits a unique stable fixed point (red dot in Fig 2A) together with a pair of unstable fixed points (orange, green). In the absence of external inputs or internal finite-size fluctuations, the system will remain in the stable, low-activity state forever. This state, however, is excitable: Fluctuations can lead to rapid, transient excursions of the neural trajectory, when the system is kicked across a separatrix (red-dashed curve = stable manifold emanating from the unstable saddle point (orange diamond)), see the blue traces in Fig 2A. During an excursion along the unstable manifold of the saddle point (orange-dotted), the input potential $h(t)$, and with it the population firing rate $f(h)$, rapidly increases, which corresponds to a short synchronized burst of activity. The increased firing of spikes leads to a strong suppression of the depression variable $x$, which in turn pulls the firing rate down. Once the depression variable $x(t)$ has recovered sufficiently, finite-size fluctuations can again trigger a synchronized burst of activity (Fig 2B and 2C). Such bursts of activity, called population spikes, have been studied theoretically in the context of STP [44, 58–60] and have also been observed experimentally [60, 61].

As predicted by the mesoscopic model, the corresponding microscopic network model also exhibits population spikes with a similar rate, and this rate increases when the network size decreases (Fig 2D and 2G). Quantitatively, the mesoscopic model accurately reproduces the statistics of population spikes in the microscopic model: There is an excellent match between the power spectra (Fig 2E and 2H) and between the distributions of the inter-population spike intervals (Fig 2F and 2I). The slight deviations of the diffusion model Eq (3) with Gaussian noise (orange, dashed traces) for a network of $N = 30$ neurons disappear for a network of $N = 200$ neurons: as expected, the diffusion approximation becomes better with increased network size. Remarkably, the jump-diffusion model Eq (4) with hybrid noise (both Gaussian and Poisson; blue traces) perfectly matches the microscopic network dynamics even when $N = 30$.

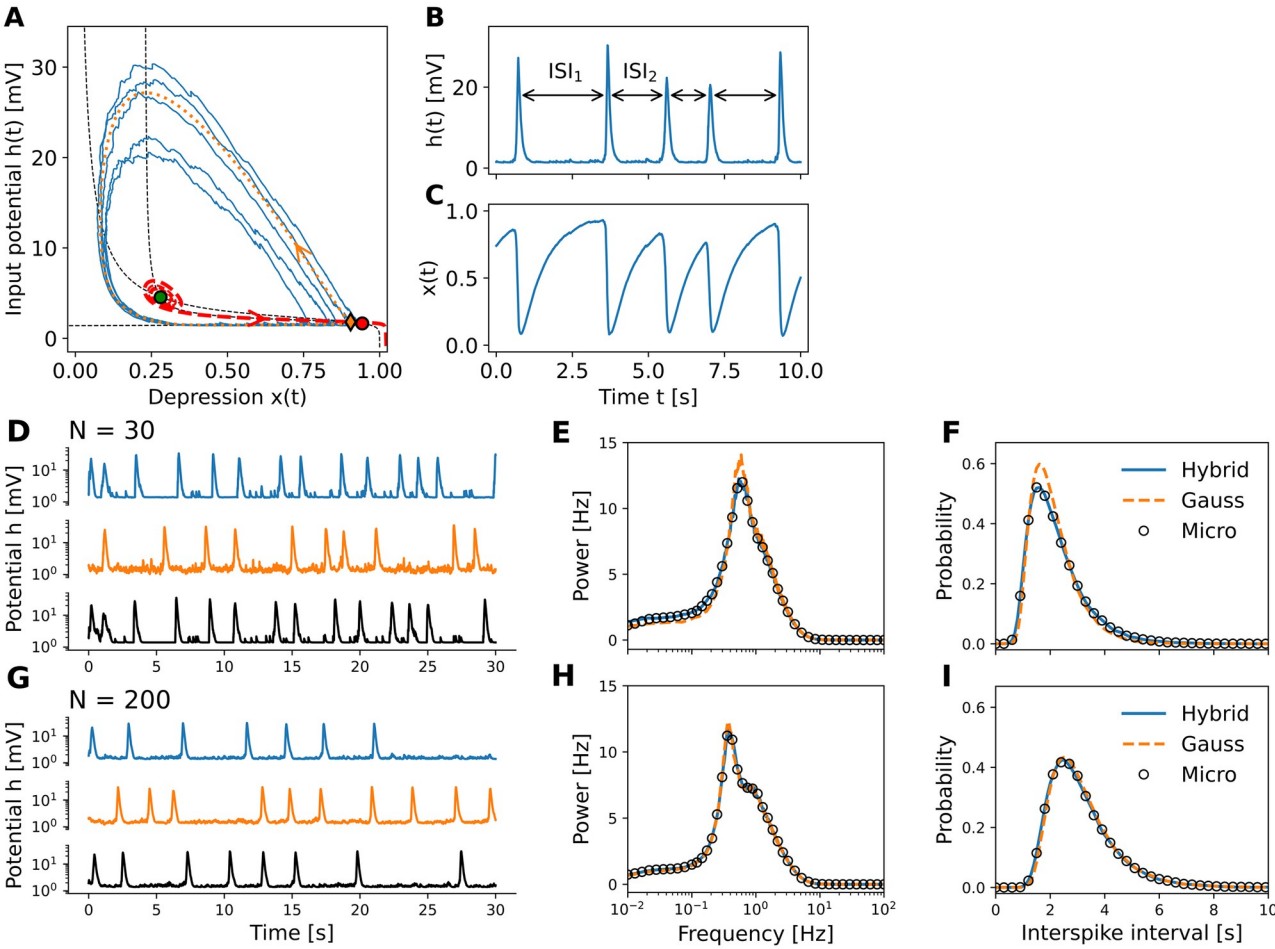

**Fig 2. Population spikes in excitatory populations of finite size.** (A) Phase-plane analysis of the macroscopic model (Eq (3) for $N \to \infty$) reveals the backbone of the metastable dynamics due to the proximity of a separatrix (red-dashed) near the unique stable fixed point (red dot = cross-section of the black-dashed nullclines). Trajectories (blue) of the mesoscopic model reproduce population spikes by following the unstable manifold (orange dotted line) of the saddle fixed point (orange diamond). Population spikes have variable amplitude and inter-population spike intervals (ISI), see also (B,C). (D) The mesoscopic models with hybrid noise (jump-diffusion model; blue) and Gaussian noise (diffusion model; orange) accurately capture finite-size fluctuations in the input potential $h$—note the logarithmic y-scale—and population spikes of the microscopic network dynamics (black) of $N = 30$ neurons. (E) Power spectra of the input potential $h$ and (F) ISI distributions coincide for all three models. (G-H) same as (D-F) for $N = 200$. Statistics are for simulations of length $T_{sim} = 100'000s$. Model parameters can be found in Table 1.

Given the close correspondence between microscopic and mesoscopic models, the mechanism of excitable dynamics driven by finite-size noise found above for the mesoscopic model thus explains the emergence of population spikes in the microscopic network model. Note that the population spikes are endogenously generated without the need for external (noisy) inputs as in previous microscopic [44, 58] and mesoscopic [44, 60] models.

**Up-Down dynamics in an excitatory population.**   In a second example, we change the model parameters slightly so that our system now exhibits two co-existing stable fixed points: a high-activity "Up" state and a low-activity "Down" state. In the macroscopic model Eq (5), which corresponds to the one studied in [16] in the absence of noise, only one of the two states can be realized depending on the initial conditions. In the mesoscopic models, Eqs (3) and (4), however, finite-size fluctuations lead to irregular transitions between Up and Down states. An exemplary stochastic trajectory in Fig 3A starts close to the Down state, but soon gets kicked across the separatrix (red-dashed stable manifold of the (orange) saddle fixed point), from

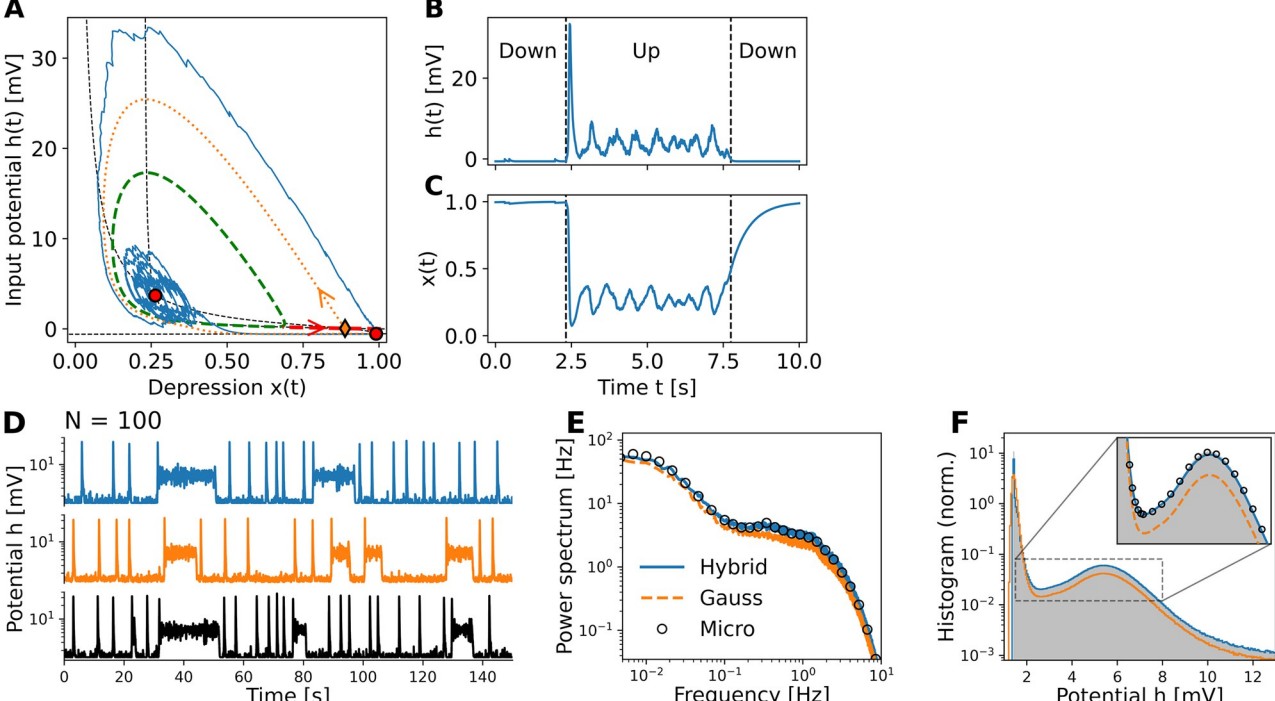

**Fig 3. Up-down dynamics due to finite-size fluctuations.** Mesoscopic model reproduces noisy bistable population dynamics. (A) Phase-plane analysis of macroscopic dynamics (Eq (3) for $N \to \infty$) reveals two stable fixed points (red): a high-activity focus representing the Up, and the low-activity node the Down state of the system. From the saddle fixed point (orange diamond), an unstable (orange dotted line) and a stable manifold (red dashed line) emerge. The latter acts as a separatrix—trajectories (blue curve) starting from above make an excursion around the unstable limit cycle (green dashed) and converge towards the down state. Finite-size fluctuations can make the trajectory cross the limit cycle into the basin of attraction of the Up state. (B, C) Stochastic trajectory of the mesoscopic dynamics (3) with $N = 100$ transitioning between Down and Up states. (D) The mesoscopic models with hybrid noise (jump-diffusion model; blue) and Gaussian noise (diffusion model; orange) qualitatively capture Up-Down-dynamics of the microscopic network (black). (E) Power spectrum and (F) histogram of input potential $h$ over simulation of length $T_{\text{sim}} = 100'000$s. Model parameters can be found in Table 1.

where it follows the (orange-dotted) unstable manifold and undergoes a sharp excursion in phase-space, resembling a population spike as described in the foregoing section. On its way back to the stable Down state, the trajectory approaches the unstable limit cycle (green dashed) that acts as the boundary of the basin of attraction of the Up state. Finite-size fluctuations can induce attractor hopping: from the low-activity node (Down state), the trajectory can cross the basin boundary and starts spiraling into the high-activity focus (Up state), until it crosses the basin boundary again and converges towards the low-activity node (Down state), see also Fig 3B and 3C. The seemingly ongoing oscillations in the Up state are a pure finite-size effect, which will be damped out in the macroscopic model. As an aside, the frequency of the oscillations in the Up state coincides with the imaginary part of the eigenvalue of the high-activity focus, cf. [46].

To assess the accuracy of our mesoscopic description of this finite-size induced metastable regime, we performed extensive simulations and compared them to the microscopic network Eq (1). In Fig 3D, we show exemplary time series of the network dynamics for $N = 100$ neurons of the jump-diffusion model Eq (4) with hybrid noise (blue), of the diffusion model Eq (3) with Gaussian noise (orange) and of the microscopic model (black). Qualitatively, there is an excellent agreement between micro- and mesoscopic simulations. However, closer inspection of the time series reveal that the Up states in the diffusion model are, on average, of shorter duration than in the microscopic and the jump-diffusion model. This slight shortcoming of

the diffusion model also becomes evident when looking at the power spectrum and the bimodal distribution of the input potential $h(t)$ computed over a long simulation of $T_{\text{sim}} = 100'000s$ (Fig 3E and 3F, respectively). The jump-diffusion model perfectly captures the full statistics of the microscopic network, but the diffusion model slightly underestimates the time spent in the Up states (see the zoom in Fig 3F), which also manifests in small deviations of the power spectrum.

## Mesoscopic model for hippocampal replays

We now turn to a more complex biological example for metastability in neural circuits: the spontaneous replay of activity sequences across hippocampal place cells [62, 63]. Sequential activation patterns of place cells have been widely observed in experiments when an animal explores its environment [64–67] and have been related to neural representations of animal trajectories, which may subserve navigation and spatial learning [68–71]. Once an internal representation, or map, unique to one environment [72] is formed, it can later be replayed spontaneously in the absence of sensory input—a feature that is believed to contribute to memory consolidation and retrieval [73–75] as well as to route planning [76, 77]. Spontaneous replay occurs within so-called sharp waves (SW)—i.e. population bursts of co-active pyramidal cells that give rise to large amplitude extracellular waves— during quiet wakefulness [78–80] and sleep [81, 82], typically has a much faster, compressed time scale [78, 82, 83] and the replayed trajectories can either be in the originally experienced order or backwards [84]. Spontaneous replay events appear and disappear abruptly and repeatedly, and can therefore be regarded as metastable states separated by states of low activity. In the following, we extend our mesoscopic theory to a ring-attractor model consisting of multiple neuronal populations in order to provide a mechanistic description of hippocampal replay with a direct link to microscopic networks of spiking neurons.

**Microscopic and mesoscopic multi-population model of place cells.** We aim for a mesoscopic description of place cells in area CA3 of the hippocampus. Following Romani and Tsodyks [21], we consider a network of neuronal populations, where each population is a group of neurons with highly overlapping place fields. We assume that the full map of the environment is covered by in total $M$ populations each containing $N$ neurons. The activity of individual neurons $j \in \{1, \ldots, N\}$ in a given population $\alpha \in \{1, \ldots, M\}$ is described by the spike trains $s_j^\alpha(t) = \sum_k \delta(t - t_{j,k}^\alpha)$ associated with the spike times $\{t_{j,k}^\alpha\}$. As in Eq (1), we model the spike trains of neuron $j$ in population $\alpha$ as a stochastic point process with conditional intensity $f(h^\alpha(t^-))$ depending on the input potential $h^\alpha(t)$, which is identical for all neurons in population $\alpha$ under the assumption of full connectivity and homogeneous initial conditions. In our microscopic ring-attractor network model with STD, the input potentials $h^\alpha$ and the neuron-specific synaptic depression variables $x_j^\alpha(t)$ then follow the dynamics:

$$\frac{\mathrm{d}h^\alpha}{\mathrm{d}t} = \frac{\mu^\alpha - h^\alpha}{\tau} + \frac{1}{M}\sum_{\beta=1}^{M}\frac{J_{\alpha\beta}}{N}\sum_{j=1}^{N}U_0 x_j^\beta(t^-)s_j^\beta(t), \tag{7a}$$

$$\frac{\mathrm{d}x_j^\alpha}{\mathrm{d}t} = \frac{1 - x_j^\alpha}{\tau_D} - U_0 x_j^\alpha(t^-)s_j^\alpha(t). \tag{7b}$$

The input potential $h^\alpha(t)$ integrates the external input $\mu^\alpha$ (common to all neurons in population $\alpha$) and the recurrent input. The latter consists of contributions from the neurons in the same population but also from all the other populations $\beta \neq \alpha$, weighted with a synaptic strength $J_{\alpha\beta}$ that depends on the distance between the place fields of the corresponding

populations (as detailed below). The resulting recurrent connectivity of the network with weights $J_{\alpha\beta}$ is assumed to encode the internal representation of one (or multiple) environment(s) that the animal has explored recently. It should be noted that the recurrent weights of an internal map can also be "learnt" via spike-timing-dependent synaptic plasticity (STDP) during active exploration of the environment, see, e.g., [10, 71]. Here, however, we assume for simplicity that the animal has already internalized the relevant environments and that the corresponding internal maps are hardwired (at least on the relevant time scale) within the synaptic connectivity matrix $\{J_{\alpha\beta}\}_{\alpha,\beta}$ in a Hopfield-like manner [85].

Analogously to the case of one population, we can reduce the microscopic dynamics Eq (7) to a mesoscopic model with multiple populations. Introducing the mesoscopic quantities $x^{\alpha}(t)$ and $Q^{\alpha}(t)$ that correspond to the first and second moment, respectively, of the depression variables $x_i^{\alpha}(t)$ for population $\alpha \in \{1, \ldots, M\}$ and defining the variances $y^{\alpha}(t) = Q^{\alpha}(t) - x^{\alpha}(t)^2$, we obtain the mesoscopic dynamics

$$\frac{dh^{\alpha}}{dt} = \frac{\mu^{\alpha} - h^{\alpha}}{\tau} + \frac{1}{M}\sum_{\beta=1}^{M} J_{\alpha\beta} U_0 \left[ x^{\beta} A^{\beta}(t) + \sqrt{\frac{y^{\beta}f(h^{\beta})}{N}}\xi_x^{\beta}(t) \right], \tag{8a}$$

$$\frac{dx^{\alpha}}{dt} = \frac{1 - x^{\alpha}}{\tau_D} - U_0 \left[ x^{\alpha} A^{\alpha}(t) + \sqrt{\frac{y^{\alpha}f(h^{\alpha})}{N}}\xi_x^{\alpha}(t) \right], \tag{8b}$$

$$\frac{dy^{\alpha}}{dt} = -\left[ \frac{2}{\tau_D} + U_0(2 - U_0)f(h^{\alpha}) \right] y^{\alpha} + (U_0 \, x^{\alpha})^2 f(h^{\alpha}), \tag{8c}$$

with Gaussian white noises $\xi_x^{\alpha}(t)$ obeying $\langle \xi_x^{\alpha}(t) \rangle = 0$ and $\langle \xi_x^{\alpha}(t)\xi_x^{\beta}(s) \rangle = \delta_{\alpha,\beta}\delta(t - s)$ for all $\alpha$, $\beta \in \{1, \ldots, M\}$. The activity $A^{\alpha}(t)$ of population $\alpha = 1, \ldots, M$ is given by

$$A^{\alpha}(t) = \frac{1}{N}\frac{dn^{\alpha}(t)}{dt} = \frac{1}{N}\sum_k \delta(t - t_k^{\alpha}), \qquad dn^{\alpha}(t) \sim \text{Pois}\left[Nf(h^{\alpha}(t^-))dt\right], \tag{8d}$$

where the points in time $\{t_k^{\alpha}\}_k$ are a realization of a point process with conditional intensity $Nf(h^{\alpha}(t^-))$ and can be interpreted as the collection of all spike times generated by population $\alpha$.

We here present only the jump-diffusion model with hybrid noise because we used this version of the model in the simulations for the following figures. However, there is also an obvious multi-population extension of the simpler diffusion model, Eq (3), see Methods, Eq (36). In some parameter regimes, the simpler diffusion model already faithfully reproduces microscopic simulations and is sufficient to study the mechanisms of recent experimental observations. In other parameter regimes, as considered here, the jump-diffusion model Eq (8) with hybrid noise captures finite-size induced metastable dynamics more accurately.

**A circular environment.**   We first consider a single circular environment and assume that, after exploration, the animal has formed an internal representation (map) of the corresponding environment, which is encoded in the synaptic connectivity of the place cells in hippocampal area CA3. Accordingly, we assign to all neurons in population $\alpha \in \{1, \ldots, M\}$ a place field at angle $\theta_{\alpha} = 2\pi\alpha/M$, so that the place field locations are equally spaced on a ring, see Fig 4A. The synaptic strength $J_{\alpha\beta}$ depends on the distance between locations according to

$$J_{\alpha\beta} = J_1 \, \cos(\theta_{\alpha} - \theta_{\beta}) - J_0 = J_1 \, \cos(2\pi(\alpha - \beta)/M) - J_0, \tag{9}$$

where $J_1$ scales the strength of map-specific interactions and $J_0$ corresponds to uniform feedback inhibition. For $J_1 > J_0 \geq 0$, populations with adjacent place fields excite each other,

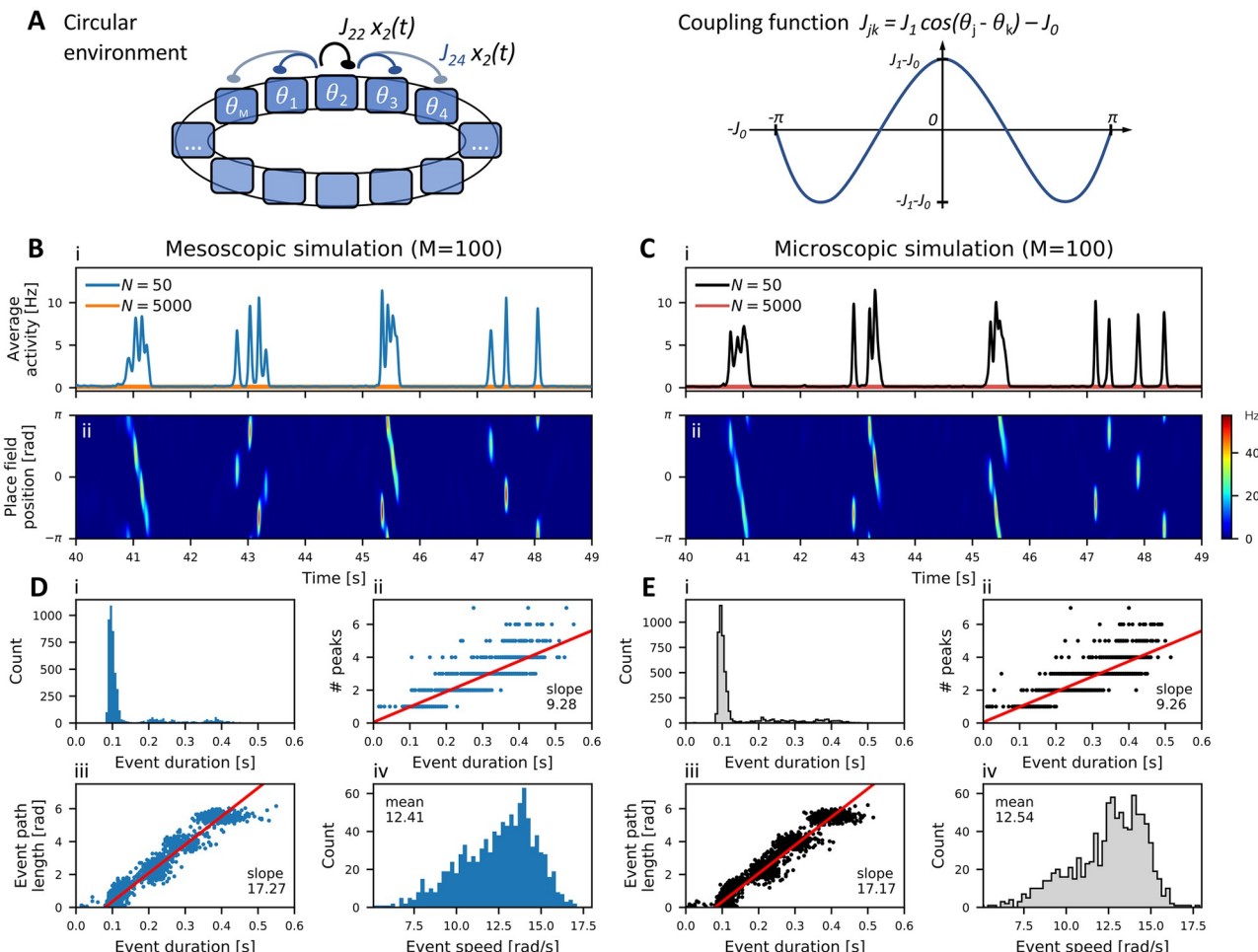

**Fig 4. Hippocampal replay in micro- and mesoscopic ring-attractor network model.** (A) Ring-attractor model of $M$ population units of $N$ LNP spiking neurons with STD. Synaptic weights $J_{\alpha,\beta}$ are excitatory for units with nearby place field positions $\theta_\alpha$ and inhibitory at longer distances, see the coupling function on the right. (B) Mesoscopic and (C) microscopic network simulations reveal (i) spontaneous bursts of the averaged activity, resembling SWs, during which replay patterns evolve as (ii) metastable traveling waves, or nonlocal replay events (NLE), along the circular environment—the expected activity $r_j = f(h_j)$ at location $\theta_j = 2\pi j/M$, $j = 1, \ldots, M$ is color-coded. Statistics of the (D) mesoscopic and (E) microscopic simulations perfectly match each other with respect to: (i) the distribution of event duration, (ii) the correlation between the number of peaks per burst and its duration, (iii) the correlation between the length of the traveled path during an event and its duration (the red curves in panels (ii,iii) are linear regression curves), as well as (iv) the distribution of average bump speed during an event, computed from events with more than one peak; see Methods for more details, Table 1 for model parameters and Table 2 for simulation results.

whereas populations with place fields far apart from each other are inhibitory. This form of symmetric interaction is known to generate spatially coherent activity, leading to so-called bump attractors [86–89]. STD has been shown to destabilize stationary bumps that move around the environment [90], creating burst-like nonlocal traveling wave events (NLEs) that resemble hippocampal replay patterns on a population level.

In our network simulations, we consider $M = 100$ populations around the ring with $N = 50$ neurons each. The external input is homogeneous, i.e. $\mu^\alpha = \mu$ for all $\alpha = 1, \ldots, M$. As shown in Fig 4Bi, the network spontaneously generates a burst of elevated activity with one or more peaks, that lasts up to a few hundred milliseconds and is then terminated due to STD. Another burst is generated only after a recovery period—the so-called interburst interval (IBI). The intermittently elevated activity on the ring is strongly localized because of the synaptic

connectivity. STD makes the localized bump move to neighboring place field locations and, thus, gives rise to an NLE, Fig 4Bii; we define an NLE (= nonlocal replay event) as a burst of the averaged activity with more than one peak, see also Methods. As the elevated activity locally alters the spatial profile of the slow synaptic depression variable, highly irregular activity patterns emerge with bursts that start at seemingly random locations, travel in either backward or forward direction, and vary both in duration and distance (note, however, that a single wave of activity never travels further than once around the circle). This type of metastable dynamics strongly resembles the behavior observed by Romani and Tsodyks in their deterministic firing rate model [21], hereafter referred to as the RT model, but with the important difference that here the emergence of metastability is a finite-size effect—increasing the population size from $N = 50$ to $N = 5000$ renders any initiation of burst-like activity impossible (see the orange curve in Fig 4Bi). Put differently, our model reveals a novel dynamical regime, in which metastable burst states are fluctuation driven, in contrast to the RT model, where bursts are induced deterministically when depression slowly abates (for a similar mechanism of noise-induced traveling waves, see [91]).

Comparing the simulations of our mesoscopic model with the microscopic network, we find an excellent agreement both from a qualitative (Fig 4B and 4C) and from a quantitative perspective (Fig 4D and 4E), where we follow the statistical analysis of [21], see also Methods. For comparison, we considered two deterministic models with fatigue-induced bursts: (i) the original RT model [21], and (ii) our model with $N \to \infty$ and slightly increased external drive ($\mu = -0.9$ instead of $\mu = -1.4$). This second model, which will be referred to as the *macroscopic* model in the following, essentially reproduces the dynamics of the original RT model [21]. We included this second model in the comparison (see also Fig A in S1 Text) because the macroscopic limit $N \to \infty$ of our mesoscopic model yields slightly different model equations compared to the heuristic RT model. A closer inspection of the statistical properties of NLEs and IBIs reveals that mesoscopic and microscopic simulations not only match almost perfectly, but the fluctuation-induced bursts may also have more biologically realistic properties than the fatigue-induced regime. For example, experiments in rodents exposed to long linear tracks [83, 92] had an experimentally estimated number of ∼ 10 SWs/s. Assuming that each peak in the average activity corresponds to one SW, the micro- and mesoscopic models closely match the experimental observations with 9.3 SWs/s in contrast to less than 8 SWs/s in the macroscopic and original RT model, see Methods for more details.

Furthermore, our fluctuation-driven model reveals larger temporal variability with a unimodal IBI distribution (Fig 5A) and low serial correlations of the event speeds (Fig 5B). In marked contrast, in the macroscopic and RT models of fatigue-induced bursts, the IBI distributions are bimodal, and clearly not exponentially distributed as observed experimentally [62, 93, 94] and expected for a Poisson process. By contrast, our fluctuation-driven model shows the tendency towards exponential IBI distributions with longer tails, showing in particular a larger mean (0.65 s vs. 0.29 s) and a higher coefficient of variation (CV = 0.85 vs. 0.79). Moreover, the macroscopic and RT models exhibit strong correlations between forward and backward replay events as seen in the serial correlations of the event speed (Fig 5Bii). The alternating structure of the serial correlation coefficient with strong anti-correlations at lag 1 means that forward and backward motion alternate almost perfectly. In contrast, the motion directions in the sequence of NLEs in our fluctuation-driven model are almost uncorrelated (Fig 5Bi). Another difference to the deterministic models is that the onset location of the fluctuation-induced NLEs is independent of the offset location of the preceding event. By contrast, in the deterministic fatigue-induced (macroscopic and RT) models, the activity bursts start at the location where the slow depression variable has had most time to recover, leading to more regular event patterns (see also [21] and Fig A in S1 Text).

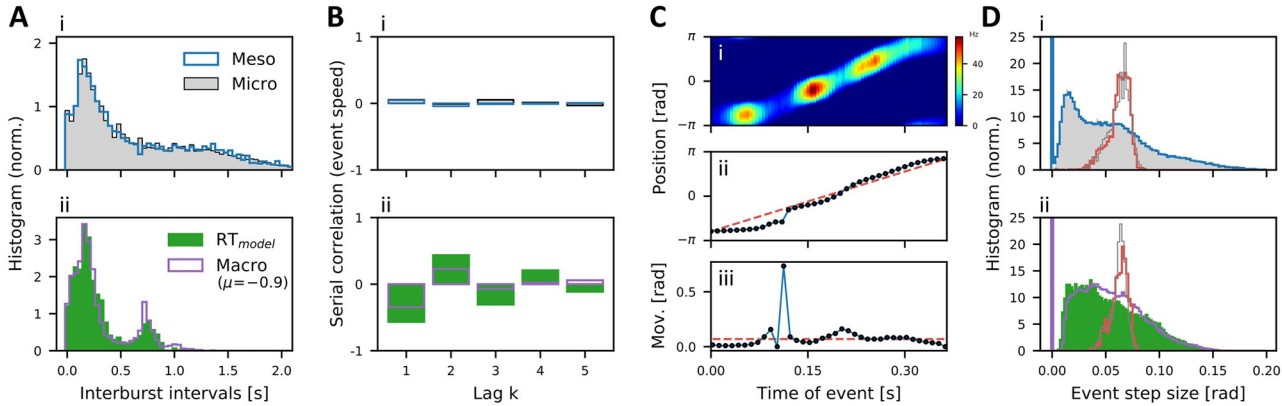

**Fig 5. Comparison of fluctuation-induced and depression-induced hippocampal replay dynamics.** (A) Interburst-interval distributions and (B) serial correlations of the event speeds of consecutive NLEs for (i) the meso- and microscopic models and (ii) the deterministic models (green: original Romani-Tsodyks model [21], purple: macroscopic model in the fatigue-driven regime obtained by setting $N^\alpha \to \infty$ and $\mu = -0.9$ (instead of $\mu = -1.4$)). (C) On shorter time scales, all models capture the discontinuous nature of the replayed trajectory: (i) Single NLE of the mesoscopic model. (ii) Place field positions decoded using PVA in bins of 50ms (black dots). This "replayed trajectory" deviates from a straight line (red-dashed) corresponding to a hypothetical uniform motion. (iii) Increments of the movement trajectory exhibit strongly irregular movement features (black dots) in contrast to the constant increments expected for a straight line (red-dashed). (D) (i): The distributions of reconstructed step-sizes of the meso- (blue) and microscopic models (gray histogram) coincide and strongly deviate from the narrow distributions of average step sizes (computed for each NLE by fitting straight lines to individual movement trajectories) for the meso- (red) and microscopic models (gray thin line). The average step sizes vary for different NLE's, causing the non-zero width of their distribution. (ii) Similar behavior is observed for the macroscopic model (purple; with increased external input $\mu = -1.4 \mapsto -0.9$) and the Romani-Tsodyks (RT) model (green histogram); red/gray thin lines correspond to average step sizes in the macro/RT-models, respectively. See main text and Methods for more details.

On shorter time scales, all models (micro-, meso-, macroscopic and RT) exhibit the experimentally observed discontinuous nature of replay events [95]. In Fig 5Ci, we zoom into one exemplary NLE of around 350ms, during which a metastable wave travels around the ring in backward direction. The place field activity (color coded) varies critically in location and activity. Binning activity per location in moving 50ms windows, we estimate the animal's (hypothetical) position along the ring by applying the population vector average (PVA; [96]), see Fig 5Cii. The decoded trajectory (black dots) does not follow a smooth path, but the step sizes vary irregularly in length (Fig 5Ciii). In Fig 5D, we show the step-size distributions for the four different models, featuring a large number of very short steps and a long tail of larger steps. The broad distributions significantly differ from the narrow step-size distribution that would be expected if the movement trajectory was uniform (thin lines in Fig 5D). More precisely, the latter distribution describes the variation of the average step sizes (computed for each NLE) across different NLEs, which corresponds to approximating each replay trajectory by a straight line (red-dashed curves in Fig 5C), see also [95] for more details on computing the step-size distributions.

In conclusion, the mesoscopic model Eq (8) perfectly recovers the metastable replay dynamics of the microscopic network model Eq (7). While our fluctuation-driven model is similar to the deterministic models with respect to the structure of single replay trajectories (Fig 5C and 5D), the fluctuation-driven model features unimodal IBI distributions and low serial correlations of motion directions, in marked contrast to the bimodal IBI distribution and the strong serial correlations of the deterministic fatigue-induced replay model. Thus, the IBI distribution and the sequence of motion directions provide experimentally testable predictions that may be useful to disentangle the contributions of deterministic and stochastic sources of metastable hippocampal dynamics.

**Multiple circular environments.** In a next step, we assume that an animal has internalized multiple environments. The ability to code for spatial locations in multiple environments is considered one of the hallmarks of place cell activity in the hippocampus. Experimental results have shown that rodents, when exposed to two distinct environments of similar shape, most place cells are active in only one environment. However, a few place cells are active in both environments but they typically exhibit place fields at different spatial locations, which is referred to as global remapping [97, 98]. Replay events can then be observed in both neural maps corresponding to each of the two environments [79], see also [99, 100].

Following [21], we consider $K$ circular environments and store their respective maps within the synaptic connectivity $J_{\alpha,\beta}$ of the network model Eq (7). To this end, we endow each population $\alpha$ with a binary vector of selectivities for these $K$ environments, $\zeta_\alpha = (\zeta_\alpha^1, \ldots, \zeta_\alpha^K) \in \{0, 1\}^K$, where $\zeta_\alpha^k = 1$ indicates that the neurons in population $\alpha$ are selective for environment $k \in \{1, \ldots, K\}$ (i.e., the neurons contribute to the encoding of this environment) [21, 101]. Otherwise, if $\zeta_\alpha^k = 0$, population $\alpha$ is not selective for environment $k$. Selectivity to particular environments is assigned randomly, but with the constraint that $\sum_{\alpha=1}^M \zeta_\alpha^k = fM$ with $f \in [0, 1]$, i.e. exactly $fM$ populations are selective for environment $k$. Furthermore, we introduce place field locations $\theta_{\tilde{\alpha}}^k = 2\pi\tilde{\alpha}/(fM)$ for each environment $k$ and randomly assign a unique place field angle to each of the $fM$ populations $\tilde{\alpha} \in \{1, \ldots, fM\}$ selective for environment $k$. We then define the synaptic weights as

$$J_{\alpha\beta} = \frac{1}{f}\sum_{\mu=1}^K J_1 \zeta_\alpha^k \zeta_\beta^k \, \cos(\theta_\alpha^k - \theta_\beta^k) - J_0, \tag{10}$$

where map-specific interactions of strength $J_1$ only occur within environments, and $J_0$ represents uniform feedback inhibition as before [21]. In our simulations, we use $K = 3$, $f = 0.3$ and $M = 300$, and find parameters of synaptic strength $J_0$ and $J_1$ as well as the homogeneous external input $\mu$ such that replay events are again a pure finite-size effect: When increasing the population size from $N = 50$ to $N = 500$, the network remains in a quiescent state and bursts of elevated activity no longer occur in our simulation.

As can be seen in Fig 6, bursts of elevated activity occur spontaneously and with high temporal variability. During these nonlocal replay events (NLE) a metastable traveling wave state is generated randomly in one of the three stored environments; activity in the other environments is suppressed due to global inhibition. As expected, our meso- and microscopic network simulations show qualitatively very similar behavior (cf. Fig 6A and 6B). The metastable dynamics exhibit high variability with respect to the duration of NLEs and the interburst intervals (Fig 6E), the length of the traveled path during a NLE within the active environment, as well as the order of environment activation.

In more detail, we statistically analyzed the patterns of sequential activations. For instance, in Fig 6A the order of environment activation reads 313123122312321. To quantify whether replay of distinct environments, i.e. their order of activation, is random or correlated, we first computed the transition probabilities between environments. While the stochastic (meso- and microscopic) models did not show strong preference for any environment transition, in the deterministic (macroscopic and RT) models, preferred transitions were clearly visible (Fig 6C). Moreover, we checked whether particular sequences of recalled environments are more probable than others, which may point at some (hidden) deterministic origin of sequence generation and recall. In order to avoid spurious deterministic effects that may be inherited from asymmetries in the synaptic weights, we constructed the selectivity vectors $\zeta_\alpha$ symmetrically and guaranteed that bursts within the environments were equally distributed, see Table 3 in

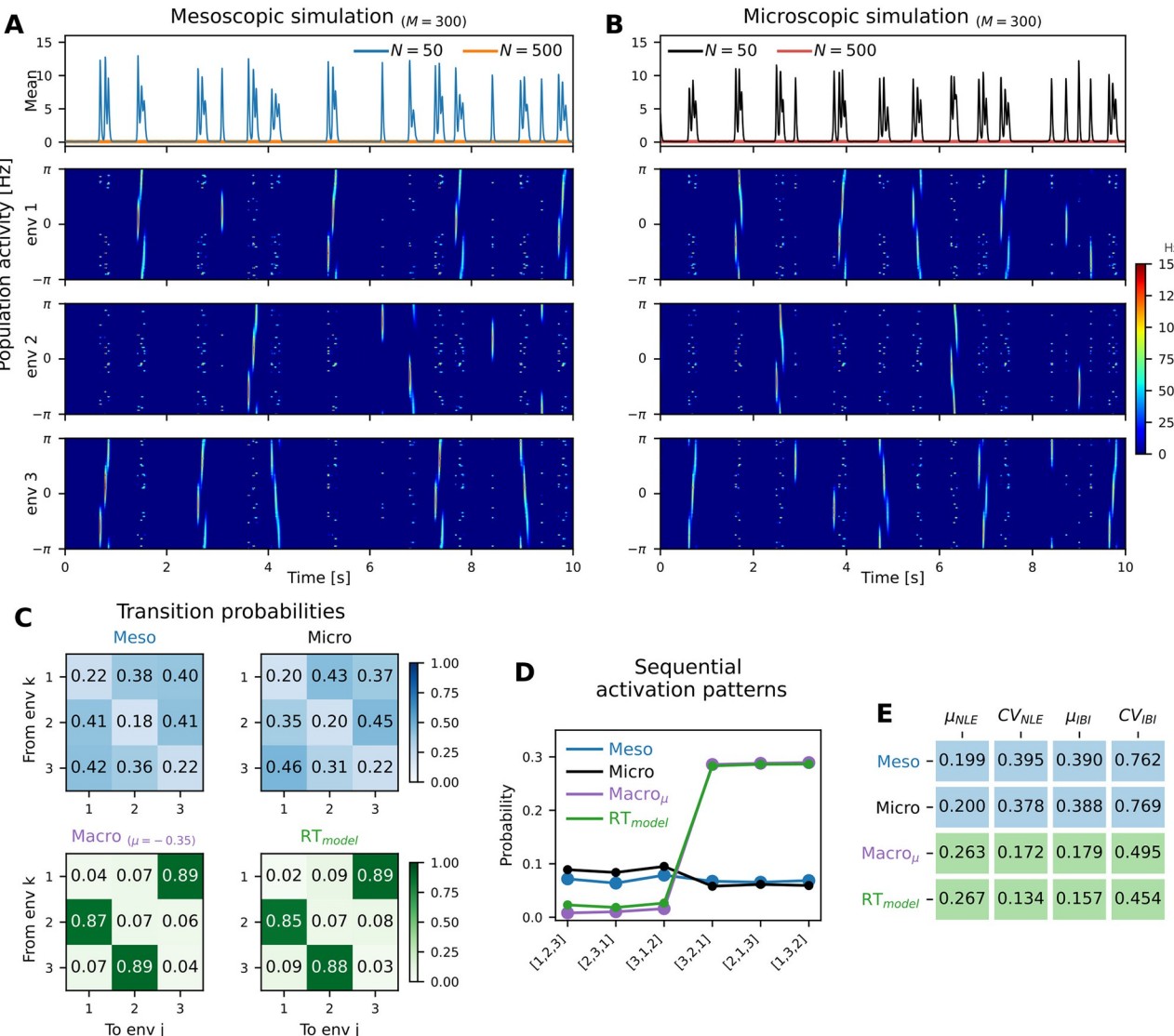

**Fig 6. Spontaneous replay switches between multiple environments.** In the (A) mesoscopic and (B) microscopic ring-attractor network storing multiple environments, metastable replay dynamics spontaneously emerge due to finite-size fluctuations when decreasing the population size from $N = 500$ (orange/red in panels i) to $N = 50$ (blue/black) per unit. Nonlocal replay events (NLEs) occur randomly in exclusively one of three environments, while activity in the respective other two is suppressed. The resulting activation sequences of replayed environments—in (A) the activation sequence reads 313123122312321—are analyzed with respect to (C) the transition probabilities between subsequently active environments and (D) sequential activation patterns. In the meso- and microscopic models, transitions from environment $k$ to $j$ are equally likely for all pairs $(k, j) \in \{1, 2, 3\}^2$. But the deterministic (macro and RT) models show a clear preference for transitions $1 \to 3 \to 2 \to 1$, which is also apparent in the high probability of the corresponding subsequences of three distinct, subsequently active environments. (E) Larger heterogeneity with respect to NLE duration and interburst intervals (IBI), as assessed by the respective means $\mu$ and coefficients of variation $CV$, further distinguishes the more variable metastable regime of the micro-/mesoscopic vis-à-vis macroscopic/deterministic models. Model parameters can be found in Table 1.

the Methods section. The deterministic (macroscopic and RT) models, nonetheless, exhibited a strong preference for specific order of environment activation ($3 \to 2 \to 1 \to 3$ in the example of Fig 6). By contrast, the stochastic models did not show any preference for a particular order, Fig 6D, which underlines once more the strong variability in the finite-size induced metastable dynamical regime of hippocampal replay.

## Discussion

To better understand the mechanisms of emerging collective dynamics and metastability in neural networks, low-dimensional mean-field models have become indispensable in theoretical, computational and systems neuroscience. In this paper, we have proposed a novel mesoscopic mean-field model for networks of spiking neurons with short-term synaptic plasticity. This mesoscopic model readily allows for systematically analyzing the effect of finite-size fluctuations on metastable dynamics. Following a bottom-up approach, we have derived simple stochastic differential equations for networks consisting of a finite number of Linear-Nonlinear Poisson (LNP) spiking neurons with dynamic synapses undergoing short-term synaptic plasticity (STP). The mesoscopic model comes in two variants: First, a jump-diffusion model captures the network dynamics of only a few neurons with high accuracy thanks to a hybrid formulation of the finite-size-induced fluctuations, which takes the shot-noise properties of the spike-train inputs into account. Second, using a diffusion approximation, we obtained an even simpler diffusion model for pure short-term depression, whose accuracy naturally improves with increasing network size. Noteworthy, its accuracy also depends on the dynamical regime under investigation, e.g., when the skewness of the shot noise critically affects the transitions to the Up-states. Nonetheless, as we showed above, the mesoscopic diffusion model is able to capture the microscopic network dynamics formidably well, allowing us to uncover finite-size induced population spikes, spontaneous transitions between Up and Down states, and a novel dynamical regime of quasi-traveling waves as a putative mechanism for fluctuation-driven hippocampal replay.

### Modeling population spikes and Up-Down dynamics

In the modeling literature, theoretical models of metastability are typically based on an interplay between the network's tendency to posit itself in a self-excitable dynamical regime and some fatigue mechanism that generates activity-dependent self-inhibition in response to elevated network activity [102, 103]. Such a fatigue mechanism can be implemented in neural networks via neural spike-frequency adaptation (SFA) or via synaptic short-term depression (STD). Here, we focused on STD, but acknowledge that similar behavior can, in principle, also be achieved with SFA. One possibility of self-excitability is that a stable low-activity state of asynchronous activity is close to a Hopf bifurcation, at which it becomes destabilized in favor of stable global oscillations. In the subcritical regime, noise can promote transient departures from the fixed point, resembling populations spikes [60]. Another possibility is that the system exhibits two stable fixed points, a high-activity (Up) and a low-activity (Down) state. Switching between the states can be induced by internal or external noise. In addition, the Up state can get destabilized in a saddle-node bifurcation by an adaptive fatigue mechanism described above [18, 20, 104, 105]. These scenarios are effectively low-dimensional and, in consequence, firing rate models describing the mean population activity can successfully explain metastable dynamics. As noted in the Introduction, however, the firing rate models largely miss a clear link to microscopic circuit models. While neuronal and synaptic properties can be partly accounted for by mean-field models, incorporating biologically realistic fluctuations in such models is more difficult as it often lacks a rigorous footing. Neglecting fluctuations may explain purely deterministic, fatigue-induced metastable activity patterns with low variability. Experimentally observed metastability in the brain, however, shows larger variability suggesting fluctuation-induced metastable dynamics. Fluctuations can have manifold origins that range from external noisy (cortical or thalamic) inputs or background noise [106], via specific network connectivity topologies (random, sparse, or clustered) [107–111], heterogeneity [112], (loose or strong) balance between excitation and inhibition, up to finite-size effects [60,

113, 114], or even a combination of them [115, 116]. On the mesoscopic scale, such fluctuations are often modeled heuristically by adding noise terms to the mean-field equations *ad hoc*.

With the mesoscopic mean-field model that we have proposed here, we restricted ourselves to explaining the fluctuations observed on a mesoscopic population level that are due to a finite number of stochastic neurons. To minimize confounding factors, we considered just one excitatory population of Poisson spiking neurons, all-to-all coupling, and no external noise. Indeed, it has been shown that inhibition is not necessarily needed to generate population spikes and Up-Down transitions, neither is short-term synaptic facilitation needed; see [16, 45, 46] who used a mean-field model with STD and Gaussian white noise in the voltage dynamics. Building on their previous work and complementing that mean-field model by an additional facilitation dynamics, Holcman and co-workers more recently provided an improved, and analytically tractable, description of network bursts, Up-Down dynamics and slow oscillations that is compatible with experimental recordings on various scales [117, 118] and allows for detailed stochastic analysis [119–121]. In the Methods, we extend our mesoscopic description considering also short-term facilitation. Our mesoscopic mean-field model Eq (3) yields exactly the same deterministic dynamics of their depression(-facilitation) mean-field model in the limit $N \rightarrow \infty$, see also [122]. The important difference, however, is how to deal with noise. While Holcman, Tsodyks and co-workers rather vaguely motivated an additive Gaussian white noise term that is meant to represent the fluctuations from independent vesicular release events and/or closings and openings of voltage gated channels, we here provide an explicit and rigorous derivation of the multiplicative noise terms in our mesoscopic mean-field model. Our model can thus accurately account for the finite-size fluctuations of the microscopic network including the thereby induced heterogeneity of synaptic depression across the neurons.

Previous approaches to model finite-size effects in networks with STP included a multiplicative noise term *ad hoc* in the firing rate equation [123, 124]. By contrast, we here derived Langevin equations directly from the underlying microscopic network of finitely many LNP spiking neurons. In our bottom-up approach we explicitly take into account fluctuations due to the variability of the individual depression variables across synapses, that are typically neglected in the other approaches. Our resulting mesoscopic model remains nonetheless simple enough to allow for an efficient analysis of finite-size induced metastable dynamics in the presence of a slow fatigue mechanism in form of STD.

## Diffusion- vs. jump-diffusion model: It depends on network size and dynamical regime

For a single population of LNP-STD neurons, we showed that microscopic network simulations were accurately captured by our mesoscopic description. In general, the agreement with the jump-diffusion model Eq (4) was excellent both for the population spike and the Up-Down states dynamics. The fit between the diffusion model Eq (3) and the microscopic network became better, the larger the network size $N$ in line with the diffusion approximation requiring a rather large $N$. At least this is true concerning the population spikes dynamics (Fig 2). In the Up-Down dynamics, however, even for intermediate network sizes of $N = 100$ neurons, deviations between microscopic simulations and the diffusion model still prevailed, see inset in Fig 3F. An increase in network size hardly led to a better fit. On top of that, a larger $N$ resulted in finite-size fluctuations with smaller amplitude, which made attractor hopping between Up and Down states more difficult and less frequent, and thus required longer simulation times. The origin of the deviations can be found in the phase plane shown in Fig 3A: the unstable manifold of the saddle fixed point (orange dotted line) leads the neural trajectory

close to the basin boundary (green dashed) of the Up state only in a small region of the phase space. In this region, fluctuations need to perturb the neural trajectory in a particular direction so that it can enter (and then also remain within) the basin of attraction of the Up state. The jump-diffusion model actually recapitulates the correct fluctuations, whereas noise in the diffusion model is too diffusive. This discrepancy can already be anticipated from the $Q$- and $y$-dynamics in Eqs (3) and (4), respectively, which directly influence the finite-size fluctuations. In fact, the steady state profiles of $Q$ and $y$ predict that the fluctuations are strongest for intermediate firing rates $f(h)$ and depression levels $0 < x < 1$, that is, right in the aforementioned critical region of the phase space, where a delicate balance between Poisson and Gaussian noise is important to recover the microscopic network dynamics. Consequently, the accuracy of the simpler diffusion model Eq (3), and hence its choice over the more complex jump-diffusion model Eq (4), to describe the microscopic network dynamics depends not only on the network size $N$, but also on the dynamical regime under investigation.

In the case of the hippocampal replay dynamics in the ring-attractor model with multiple populations of LNP-STD neurons, we saw a very similar effect. For fatigue-induced hippocampal replay, the actual characteristics of the finite-size fluctuations were less important than the deterministic backbone provided by the spatio-temporal profile of the slow fatigue (depression) variable. Hence, the diffusion model was able to accurately capture the metastable replay dynamics of the microscopic ring network. By contrast, for the chosen parameter set in Fig 4, hippocampal replay patterns were finite-size induced and vanished completely for large network sizes. As such, capturing the finite-size noise properties of the microscopic network was necessary for observing the NLE dynamics. The finite-size induced hippocampal replay dynamics could be recovered by both mesoscopic model versions, albeit the jump-diffusion model yielded a better quantitative agreement.

## Network size variation resembles long-term plasticity

Our study of the Up-Down-dynamics has been guided by the work of Holcman, Tsodyks and co-workers [16, 45], who considered a firing rate model with external stochastic input as a necessary ingredient to realize the metastable dynamics and irregular transitions between Up and Down states in a network with STD, which was supported by experimental observations in [46]. The Up and Down transitions described by our mesoscopic model in Fig 3A–3C solely stem from internally generated finite-size fluctuations and no external noise is needed. In addition, the mesoscopic model can explain certain dynamical features that Holcman and Tsodyks ascribed to long-term synaptic plasticity by changing the network size. Specifically, in [16] Holcman and Tsodyks attested deviations from typical Up-Down dynamics to external stimulation (by changing the input parameter $\mu$) or to long-term synaptic plasticity (by changing the recurrent coupling strength $J$). A depolarization injection current (larger $\mu$) had a similar effect as long-term potentiation (LTP; stronger recurrent coupling $J$), which led to longer Up states. On the other hand, a hyperpolarization injection current (smaller $\mu$), or similarly long-term depression (LTD; smaller $J$), led to shorter Up states and more frequent population spikes. While our mesoscopic description predicts analogous behavior of the microscopic network when changing $\mu$ and/or $J$, it also predicts that similar effects can be realized by varying the network size $N$. Performing simulations with various network sizes between $N = 50$ and $N = 150$ while keeping the other parameters unchanged, we found that for $N = 100$, metastable activity featuring populations spikes is interspersed with Up states that last on average 10.6 s (mean taken over all Up states that last at least 1 s), see Fig 3D. For smaller networks with $N = 50$ (simulations not shown), population spikes become more frequent and Up states become shorter (mean duration $3.1s$): Over a $10'000$ s simulation, Up states followed

population spikes in 16.9% of all 2249 cases for $N = 50$, whereas for $N = 100$ this only occurred in 15.6% of all 1218 cases. Larger networks with $N = 150$, by contrast, exhibit significantly longer Up states (mean duration 41.9 s) and even less frequent population spikes (see Fig B in S1 Text). Computing the corresponding histograms of the input potential revealed only for sufficiently large network sizes a second peak at around 5.5 mV (as in Fig 3F), which coincides with the Up state in Fig 3A. Furthermore, the fluctuation-driven oscillations in the long-lasting Up states (Fig 3B and 3C for $N = 100$) become visible in the power spectrum for $N = 150$ as another peak at around 1.5 Hz emerges. This frequency corresponds to the imaginary part of the eigenvalues of the stable focus ($\lambda \approx (-1.54 + 9.24i)$ Hz hence $\text{Im}(\lambda)/(2\pi) \approx 1.5$ Hz).

We can conclude that in our theoretical model the network size critically affects the Up-Down dynamics. Recent findings in mouse primary sensory cortex show that the network size changes dynamically as the proportion of stimulus-responsive cells increases with learning [125]. Likewise, learning or, more general, context-dependence may dynamically control Up and Down states by recruiting more (or less) neurons in the respective brain area. One needs to keep in mind, however, that when varying the number $N$ of neurons in our model, we have simultaneously rescaled the synaptic weights in proportion to $1/N$. Although this specific change of parameters is useful for the theoretical analysis, a corresponding biological implementation is difficult to perceive. In any case, for a given network size and synaptic weights, our mesoscopic mean-field models Eqs (3) and (4) provide accurate descriptions of the microscopic network with short-term synaptic depression and therefore allow for an efficient analysis of how finite-size fluctuations contribute to and shape metastable dynamics.

## Modeling hippocampal replay

Given the recent success in explaining a wide range of experimental observations on hippocampal dynamics by means of firing-rate models with STD [21, 71], we extended our mesoscopic description for a single population to a ring-attractor model of spiking neurons. Aiming at a minimal spiking neuron network model that can offer unique insights in the generative mechanisms of hippocampal dynamics in area CA3—in particular those underlying sharp waves and bidirectional activity replay—, we ignored various degrees of biological plausibility on purpose, but see below for possible extensions. In contrast to previous models that incorporated more biological details [10, 126–131], our derived stochastic ring model provides a relatively simple framework for a concise and efficient analysis of hippocampal sharp waves. Moreover, we were able to overcome the notorious difficulties of accommodating both the capability to endogenously generate sequential activity [132–134] and trial-to-trial temporal variability [6, 9].

In previous approaches to model metastable sequential activation patterns, as evident in hippocampal replay, successful candidate mechanisms to account for both of the above characteristics have already been implemented with the help of firing rate models. Recanatesi et al. [135] proposed a two-area mesoscale attractor network very much in the spirit of the winnerless competition model by Seliger, Tsimring and Rabinovich [22], in which the combination of asymmetric synaptic connectivity, arising from reciprocal coupling between fast and slow systems, with stochastic synaptic efficacy is crucial for the generation of sequences. Alternatively, Romani and Tsodyks [21] proposed a fully deterministic mechanism for hippocampal replay based on short-term depression (STD) and without the need for asymmetric connectivity, see also [71]: A ring-attractor network model with symmetric synaptic connectivity with local excitation and long-range inhibition exhibits transitions between a global quiescent state and various spatially localized bump states [48, 86, 87]. STD, as a slow fatigue mechanism, destabilizes these bumps and gives rise to traveling wave states [90]. In combination with the

high-dimensional character of the network, the resulting dynamics appears effectively stochastic and spontaneously switches between quiescence and metastable traveling waves. As the approaches by Recanatesi et al. [135] and by Romani and Tsodyks [21] rely on heuristic firing-rate models, both of them suffer from the limitations already discussed in the Introduction and neither can account for a systematic investigation of finite-size effects on metastable activity. It is also unclear to what extent the deterministic Romani-Tsodyks (RT) model describes neural variability. For these reasons, we saw the need for a mesoscopic theory and derived a stochastic ring-attractor model bottom-up, thereby creating a direct link to microscopic networks of spiking neurons.

We first corroborated the findings of [21, 71] as we recovered, in the limit of infinitely many neurons per population, an analogous deterministic regime of spontaneously emerging hippocampal replay patterns as in the RT model [21], see Fig A in S1 Text. Given the similarity between our macroscopic model and the RT model, we subsequently probed our mesoscopic ring-attractor models by changing first the network size and then also by tuning the strength of the common input. In the regime of fatigue-induced hippocampal replay, already the diffusion model with Gaussian noise proved an accurate description of the microscopic network dynamics with finite population sizes, see Table A in S1 Text. When decreasing the common input $\mu = -0.9 \mapsto -1.4$, the microscopic network simulations featured finite-size induced hippocampal replay dynamics for small population sizes $N$ that were accurately recovered by the mesoscopic jump-diffusion ring network model Eq (8) with hybrid noise, see Fig 4. The simpler diffusion model also captured these metastable dynamics but the quantitative match was not as good as in the jump-diffusion model, cf. Table B in S1 Text. Importantly, the macroscopic model could not retrieve the NLE regimes at all and remained in the globally quiescent state. The good agreement between the ring network simulations for single and multiple environments using either the microscopic or the mesoscopic ring network models, makes us confident that our mesoscopic description readily allows for capturing realistic hippocampal dynamics not only on periodic tracks—which we considered here for simplicity and as a proof of concept—but also on linear tracks, in T-maze environments, and planar (and higher dimensional) fields; we leave these extensions for future work as well as the formation of theta sequences and phase precession [21, 71, 136], see also [137, 138].

Recent experiments highlight the importance of stochastic models of hippocampal replay that can explain (i) replay patterns that resemble Brownian diffusion [139] or even super-diffusion [140], (ii) interburst intervals that exhibit significant variability [62, 93, 94], and (iii) replay episodes that do not always draw a smooth continuous path, but often follow a jumpy, discontinuous trajectory [95, 141]. Our mesoscopic description can be a useful step towards such a stochastic modeling framework. It may allow for a mechanistic understanding of the neuronal, synaptic and network constituents for generating spontaneous hippocampal activity, which are critical for memory consolidation, recall and spatial working memory, navigational planning, as well as reward-based learning [63, 75, 77, 80].

### Predictions and possible functional roles of variable hippocampal dynamics

Our mesoscopic description opens a new perspective on the variability of hippocampal dynamics. Using the mesoscopic model, we uncovered a novel, finite-sized induced metastable regime of hippocampal replay (see also [91] for a similar mechanism of noise-induced traveling waves). At first glance, these fluctuation-induced quasi-traveling waves are similar in nature to those in [21]. On time-scales of a few hundred milliseconds, both the deterministic RT model and our mesoscopic model exhibit spontaneously emerging bursts of activity that

resemble a nonlocal replay event. As a single replay event unfolds, we showed that both models feature discontinuous replay trajectories—consistent with experimental observations [95, 140, 141], which also underlines the biological plausibility of the ring-attractor assumption. On longer time-scales, however, our mesoscopic model allows for a much richer repertoire of replay dynamics. We found that the mesoscopic ring-attractor network can exhibit significantly larger variability for finite-sized induced replay dynamics than for deterministic, fatigue-induced dynamical regimes. This variability manifests in the spatio-temporally irregular succession of replay events (Figs 4 and 5 and Fig A in S1 Text) and could be tested experimentally, following, e.g., [62, 93, 94]. Consequently, our results insinuate that different replay dynamics can have distinct dynamical origins. In particular, replay dynamics in rodents are reportedly different during awake rest versus sleep [140, 142], possibly relating to fluctuation- versus fatigue-induced metastability.

## Replay in brains and machines

Our findings may also be relevant for human neuroscience, where a paradigm shift to decode cognition rather from off-task, than from task-based, neural activity seems to be imminent [143]. Hippocampal replay in rodents is a prime example for a "representation-rich" approach to spontaneous neural activity by uncovering the temporal structure of task-related representations. In understanding how the temporal dynamics of a particular neural activity pattern (e.g., a nonlocal traveling wave) unfolds, researchers could shed light on various cognitive functions that are subserved by spontaneous neural activity, including memory, learning, and decision-making [77, 144]. Recent technical advances in human neuroimaging have inspired "human replay" studies that investigate spontaneous task-related neural reactivations [145–147], which bears strong resemblance to rodent replay. Instead of the spontaneous recall of an environment map in rodent hippocampal replay, the focus lies now on reactivation of a more abstract "cognitive map" of task space. As the associated cognitive processes include memory retrieval, planning and inference, and thus lie at the heart of sophisticated model-based reasoning, our results can also be regarded as a proof-of-concept for the model-based representation-rich approach advocated in [143] when re-interpreting the environmental maps stored in synaptic connectivities as cognitive maps. Intriguingly, also in human replay studies, replay can occur in forward and backward direction, with putative functional roles for spatial and non-spatial learning [148, 149]. In particular, Liu and co-workers suggest that nonlocal backward replay may serve as a neural mechanism for model-based reinforcement learning [149]. A comprehensive view about the different roles the wide variety of replay dynamics may subserve, however, remains elusive.

Insights from machine learning, where replay is commonly implemented in artificial agents, may help to find answers about the putative computational functions [150, 151]. "Experience replay" was already introduced as a reinforcement learning technique in the early 90s [152] and is nowadays a crucial ingredient in building human-level intelligence in deep neural networks [153, 154]. Note also that in reinforcement learning, a 'model' has a similar meaning to the notion of a 'cognitive map', which thus naturally bridges the gap from (human) cognition to artificial intelligence [155]. Nonetheless, research on replay in neuroscience and machine learning has progressed largely in parallel, so that insights from the latter can also inform future neuroscientific studies. An outstanding problem in the field of deep learning is the catastrophic forgetting problem in online learning [151, 156, 157]. This problem is due to the fact that during online learning, data is not guaranteed to be independent and identically distributed (*i.i.d.*), which is a challenge for standard optimization methods. Replay-like methods are used to overcome this problem, but current replay implementations are

computationally expensive to deploy. By contrast, uniform sampling of past experiences has proven to be remarkably efficient both in supervised learning [158–160] and in reinforcement learning tasks [153, 161]. These insights provide some indirect, yet important evidence for the computational benefit of the stochastic replay that we have uncovered in this work and hint at an important role of the novel fluctuation-induced replay regime.

## Biological limitations and extensions

In this paper, we aimed at a minimal bottom-up population model that accounts for spiking noise, short-term synaptic plasticity and basic neuronal properties. The result can be regarded as a proof of concept that a simple nonlinear mesoscopic model, which enables the analysis of metastable dynamics in terms of network size and the aforementioned properties, can be derived from an underlying microscopic model. However, our model has several biological limitations and lacks some important features. First, neurons exhibit post-spike *refractoriness*, which is not captured by the LNP model. While the response of the instantaneous firing rate can be well reproduced by choosing the linear filter function $\kappa(t)$ and the nonlinearity $f(h)$ of the LNP model corresponding to realistic dynamical transfer functions of neurons with refractoriness [54], the temporal spike-train correlations caused by refractoriness violate the Poisson assumption of our derivation. Although the strict Poisson assumption can be relaxed to some degree [44], strong spike-train auto-correlations influence the noise properties of the mesoscopic model and hence the fluctuations of the population activities. This effect has already been described for the mesoscopic model of [44] in the case of leaky integrate-and-fire neurons with pronounced refractoriness. Because our theory is based on the previous mesoscopic model, we expect that these effects carry over to the present model. How to account for non-Poisson statistics due to refractoriness in a mesoscopic theory with STP is a challenging theoretical problem that is left for future research. We also mention that a related neuronal property is spike-triggered *adaptation*, which—similar to synaptic depression—is a slow negative feedback mechanism. Incorporating adaptation into a mesoscopic theory, either instead of or in addition to depression, is interesting for two reasons: first, it represents an alternative slow fatigue mechanisms driving metastable dynamics, and second, adaptation is an important biological feature found in many cell types [162]. A promising generalization of the present theory to adaptation could be based on the quasi-renewal approximation [163] and its extension to mesoscopic theories [26, 164].

Apart from a more realistic description of neuronal properties, also the synaptic dynamics can be improved towards important biological features. First, the synaptic conductances exhibit temporal filtering and, in a conductance-based model, they enter the voltage dynamics in a multiplicative way. Both effects are neglected in our model. At least, the synaptic filtering of the conductance dynamics is straightforward to include in our theory as shown in [44]. For an extension of the mesoscopic model to a conductance-based description of synaptic input, the interested reader is referred to the discussion of [26], see also [116, 165]. Second, the Tsodyks-Markram model considered here is a phenomenological and deterministic model of the synaptic dynamics. However, synaptic transmission is stochastic, and thus a stochastic STP model [166] would be biologically more realistic (see also discussion in [44]), but how to treat such stochasticity within a mesoscopic theory remains unclear.

A biologically difficult problem is how to account for realistic network topology. Our theory applies to networks of multiple interacting homogeneous populations. In turn, each population is a fully-connected network of neurons. However, the connectivity of biological neural networks is not fully connected and often exhibits a large degree of heterogeneity. Regarding the first issue, we have shown previously that the full connectivity model represents an effective

model that faithfully reproduces the dynamics of a non-fully-connected, random network with fixed in-degree if the synaptic weights are rescaled correspondingly [44] (see also [26] for the case of static synapses). The second issue is a principle challenge for mean-field theories as they rely on the possibility to use averages over many neurons, and hence cannot describe heterogeneous networks that are strongly affected by single neurons. However, in many cases it might be valid to subdivide the network into many small subpopulations that can be regarded as roughly homogeneous. Following this strategy, it is crucial to have a mesoscopic description of the subpopulations because the grouping of neurons with similar properties may result in small population sizes. For example, in our hippocampal network model, we have grouped place cells with highly overlapping place fields into one homogeneous subpopulation. If the number of these similarly tuned place cells is small, the mesoscopic framework can show its full strength because in this case, the jump-diffusion model still provides an accurate description (see Fig 3 for $N = 30$ neurons). Finally, we note that a basic type of network heterogeneity in biology, the separation of excitatory and inhibitory neurons (Dale's law), is not realized in our hippocampal network with "Mexican-hat"-type connectivity. However, it has been shown that such connectivity can be re-implemented in accordance with Dale's law by two layers of neurons, one excitatory and one inhibitory layer [167].

## Theoretical challenges

The diffusion model Eq (3) and its derivation based on temporal coarse-graining [56] greatly simplify our previous theoretical work [44] but both results are consistent as shown below in Methods. The simplicity of our new derivation and the remarkable agreement of the diffusion model with microscopic simulations beg the question of whether the diffusion model is the *exact* diffusion approximation [168]. The only way to give a positive answer to this question would be to propose a rigorous diffusion approximation proof (as in [169] for the case of LNP neurons without STP), which we leave as an open mathematical problem. A second open theoretical question is the convergence of the multi-population model Eq (8) to a stochastic neural field equation. The circular environment we study can be regarded as a space-discretized stochastic neural field but the exact expression of the continuous equation and its physical interpretation is unclear and will be subject of future work. Ideally, one would want to be able to prove the convergence to such an equation, as in [170] in the case without STP.

The low-dimensional mesoscopic model could be also interesting from a data-analytical perspective. Recently, Bayesian state-space models have been developed to infer replay events from spiking data [141]. Such inference does not exploit any knowledge about the dynamical mechanisms underlying the data and the likelihood function is assumed *ad hoc*. Our low-dimensional mesoscopic could provide an analytical likelihood function so as to enable improved data assimilation methods to infer replay events.

In conclusion, we have put forward a multiscale framework for systematically investigating metastable network dynamics in finite-sized networks of LNP-STD neurons using a bottom-up mesoscopic model. This model is efficient to analyze and simulate and is also versatile for incorporating more biological realism. Thanks to a unique link between the underlying microscopic network and its mesoscopic description, it becomes possible to disentangle the differential roles of neuronal, synaptic and network properties—in particular the network size—for emerging metastable brain dynamics. The mesoscopic model may also be instrumental for distinguishing between fatigue-driven and fluctuation-driven metastability because of their distinct statistical predictions—as in the case of hippocampal replay. Such predictions could be tested experimentally and reveal the dynamical origin of spontaneous neural activity.

## Methods

### Diffusion approximation for the mesoscopic dynamics with short-term depression

The microscopic dynamics of a network of $N$ LNP spiking neurons with short-term synaptic depression and an exponential linear filter (leading to leaky input integration) are given by Eq (1). To derive the diffusion model of the mesoscopic dynamics Eq (3), we focus on the mesoscopic variables $h(t)$, $x(t)$ and $Q(t)$ defined as in Eq (2):

$$h(t) := \frac{1}{N} \sum_{i=1}^{N} h_i(t), \quad x(t) := \frac{1}{N} \sum_{i=1}^{N} x_i(t) \quad \text{and} \quad Q(t) := \frac{1}{N} \sum_{i=1}^{N} x_i^2(t).$$

From Eq (1) and using the definition of $x(t)$, we get

$$\frac{dh}{dt} = \frac{\mu(t) - h}{\tau} + J U_0 \frac{1}{N} \sum_{i=1}^{N} x_i(t^-) s_i(t), \tag{11a}$$

$$\frac{dx}{dt} = \frac{1 - x}{\tau_D} - U_0 \frac{1}{N} \sum_{i=1}^{N} x_i(t^-) s_i(t). \tag{11b}$$

Note that, as mentioned in the presentation of the microscopic LNP-STD model (1), if all the $h_i$ have the same initial condition at time 0, then $h_i(t) = h_j(t)$ for all times $t \geq 0$ and, consequently, $h_i(t) = h(t)$ for all $t \geq 0$. This means that for this type of initial conditions (also used in [50]), the microscopic $h_i$ are equivalent to the mesoscopic $h$. Here, we consider only this case.

To approximate the sum $\frac{1}{N} \sum_{i=1}^{N} x_i(t^-) s_i(t)$ by a diffusion term which only involves mesoscopic variables, we follow the coarse-graining approach by Gillespie [56] for the derivation of a "chemical Langevin equation". To this end, we study the stochastic increments $\int_t^{t+\Delta t} \frac{1}{N} \sum_{i=1}^{N} x_i(\hat{t}^-) s_i(\hat{t}) d\hat{t}$, where $\Delta t > 0$ is assumed to be a *macroscopically infinitesimal* time step [56]: $\Delta t$ is small enough such that (i) the $x_i$'s can be assumed to jump at most once in the time interval $[t, t + \Delta t]$, (ii) $x_i(\tau_i^-) \approx x_i(t^-)$ if neuron $i$ has a spike at time $\tau_i \in [t, t + \Delta t]$, and (iii) $h(\hat{t}) \approx h(t^-)$ for all $\hat{t} \in [t, t + \Delta t]$; and $\Delta t$ is large enough such that many neurons spike in the time interval $[t, t + \Delta t]$. These assumptions are expected to hold if $\Delta t \ll \tau, \tau_D$ and $1 \ll N f(h(t)) \Delta t \ll N$ for all $t$. By the smallness assumption, we have

$$\int_t^{t+\Delta t} \frac{1}{N} \sum_{i=1}^{N} x_i(\hat{t}^-) s_i(\hat{t}) d\hat{t} \approx \frac{1}{N} \sum_{i=1}^{N} x_i(t^-) z_i(t), \tag{12}$$

where $\{z_i(t)\}_{i=1}^{N}$ are *i.i.d.* Bernoulli random variables with mean $f(h(t^-))\Delta t$. Conditioned on the past $\{(h_i(t'))_{t'<t}\}_{i=1}^{N}$ (where $(h_i(t'))_{t' < t} \equiv (h(t'))_{t' < t}$ by our assumption on the initial condition), all the variables $x_1(t^-), \ldots, x_N(t^-), z_1(t), \ldots, z_N(t)$ are approximately independent, and the $\{x_i(t^-)\}_{i=1}^{N}$ are approximately *i.i.d.* (all these variables become truly independent as $N \to \infty$ [50]). Hence, we use the Gaussian approximation,

$$\frac{\sum_{i=1}^{N} x_i(t^-) z_i(t) - N \mathbb{E}[x_1(t^-) z_1(t) \mid (h(t'))_{t'<t}]}{\sqrt{N}} \dot{\sim} \mathcal{N}\left(0, \text{Var}[x_1(t^-) z_1(t) \mid (h(t'))_{t'<t}]\right).$$

We now use the empirical averages Eq (2) to approximate the conditional expectation and variance:

$$
\begin{aligned}
\mathbb{E}[x_1(t^-)z_1(t) \mid (h(t'))_{t'<t}] &= \mathbb{E}[x_1(t^-) \mid (h(t'))_{t'<t}]f(h(t^-))\Delta t \approx x(t^-)f(h(t^-))\Delta t \\
\mathrm{Var}[x_1(t^-)z_1(t) \mid (h(t'))_{t'<t}] &= \mathbb{E}[x_1^2(t^-)\mid (h(t'))_{t'<t}]\,\mathbb{E}[z_1^2(t^-)\mid (h(t'))_{t'<t}] \\
&\quad - \mathbb{E}[x_1(t^-)\mid (h(t'))_{t'<t}]^2\,\mathbb{E}[z_1(t^-)\mid (h(t'))_{t'<t}]^2 \\
&\approx Q(t^-)f(h(t^-))\Delta t + O(\Delta t^2).
\end{aligned}
$$

We can now approximate the increment Eq (12) by a Gaussian:

$$
\int_t^{t+\Delta t} \frac{1}{N}\sum_{i=1}^N x_i(\hat{t}^-)s_i(\hat{t})d\hat{t} \;\dot\sim\; \mathcal{N}\left(x(t^-)f(h(t^-))\Delta t,\ \frac{Q(t^-)f(h(t^-))\Delta t}{N}\right).
$$

Taking the limit $\Delta t \to 0$, we obtain the diffusion approximation

$$
\frac{\mathrm{d}h}{\mathrm{d}t} = \frac{\mu(t)-h}{\tau} + JU_0 xf(h) + JU_0\sqrt{\frac{Q(t)f(h)}{N}}\xi(t), \tag{13a}
$$

$$
\frac{\mathrm{d}x}{\mathrm{d}t} = \frac{1-x}{\tau_D} - U_0 xf(h) - U_0\sqrt{\frac{Q(t)f(h)}{N}}\xi(t), \tag{13b}
$$

where $\xi(t)$ is a Gaussian white noise with auto-correlation function $\langle \xi(t)\xi(t')\rangle = \delta(t-t')$.

To close the system (13), we have to derive the dynamics of $Q(t)$. Going back to Eq (1d), by Itô's formula for jump processes, we have

$$
\frac{\mathrm{d}x_i^2}{\mathrm{d}t} = 2\frac{x_i - x_i^2}{\tau_D} - U_0(2-U_0)x_i^2(t^-)s_i(t).
$$

Taking the empirical average, we get

$$
\frac{\mathrm{d}Q}{\mathrm{d}t} = 2\frac{x-Q}{\tau_D} - U_0(2-U_0)\frac{1}{N}\sum_{i=1}^N x_i(t^-)^2 s_i(t). \tag{14}
$$

We could follow the same steps as before and try to obtain a diffusion approximation for Eq (14). However, the fluctuations of such a diffusion approximation would be of order $1/\sqrt{N}$ and since $Q(t)$ affects the dynamics of Eq (13) only through the term $\sqrt{Q(t)f(h)/N}\xi(t)$, the effect of the fluctuation of $Q(t)$ on $h(t)$ and $x(t)$ are of order $N^{-3/2}$ and can therefore be neglected when $N$ is large. Hence, we approximate the increments by their (approximate) expectation: for the time step $\Delta t > 0$,

$$
\int_t^{t+\Delta t} \frac{1}{N}\sum_{i=1}^N x_i(\hat{t}^-)^2 s_i(\hat{t})d\hat{t} \approx Q(t^-)f(h(t^-))\Delta t,
$$

whence,

$$
\frac{\mathrm{d}Q}{\mathrm{d}t} = 2\frac{x-Q}{\tau_D} - U_0(2-U_0)Qf(h). \tag{15}
$$

Finally, gathering Eqs (13) and (15), we obtain the mesoscopic dynamics in the form of the diffusion model Eq (3).

The fact that we are here considering the diffusion approximation (or Langevin dynamics) allows us to significantly shorten the original derivation of the mesoscopic model presented in

[44]. In particular, in the present derivation, we do not need to approximate the distribution of the $x_i(t^-)$ in Eq (12) by a Gaussian, since we only need to approximate the sum in Eq (12) by a Gaussian. Note that the arguments enabling the present derivation were already hinted at in the Section "Remarks on the approximation" and Appendix B of [44] but not put together. Both derivations lead to the same mesoscopic model, except that here, for simplicity, we neglect fluctuations of order $N^{-3/2}$ and we consider the diffusion limit; see also the Methods Section "Reduction to a pure diffusion process" below for an explicit derivation of the diffusion model from the original mesoscopic model presented in [44].

## Jump-diffusion model with synaptic depression and facilitation

Starting from our previous work [44], we can derive an improved mesoscopic model that also accounts for synaptic facilitation and the shot-noise character of the finite-size spiking noise. When allowing only for short-term depression while keeping the facilitation variable constant, the resulting mesoscopic dynamics boils down to the jump-diffusion model Eq (4) with hybrid noise. For large $N \gg 1$, the Poisson shot noise can be simplified under a diffusion approximation, which recovers the diffusion model Eq (3) derived in the previous section.

**Microscopic model with synaptic depression and facilitation.** We consider a network of LNP neurons with dynamic synapses similar to Eq (1) but now complemented with a facilitation variable $\hat{u}_i$ for each neuron $i = 1, \ldots, N$. The full synaptic dynamics corresponds to the STP model by Tsodyks and Markram [23, 36] and results in the following microscopic network model:

$$\frac{d\hat{h}_i}{dt} = \frac{\mu(t) - \hat{h}_i}{\tau} + \frac{J}{N}\sum_{j=1}^{N} \hat{u}_j(t^-)\hat{x}_j(t^-)s_j(t) \tag{16a}$$

$$\frac{d\hat{u}_j}{dt} = \frac{U_0 - \hat{u}_j}{\tau_F} + U(1 - \hat{u}_j(t^-))s_j(t) \tag{16b}$$

$$\frac{d\hat{x}_j}{dt} = \frac{1 - \hat{x}_j}{\tau_D} - \hat{u}_j(t^-)\hat{x}_j(t^-)s_j(t), \tag{16c}$$

where $s_i(t) = \sum_k \delta(t - t_k^i)$ is a point process with conditional intensity $f(\hat{h}_i(t^-))$. In Eq (16), $\tau_F$ and $\tau_D$ are the facilitation and depression time constants, respectively, and $U_0$ is the baseline utilization of synaptic resources, whereas $U$ determines the increase in the utilization of synaptic resources by a spike. As before, $\hat{u}_j(t^-)$ is a shorthand for the left limit at time $t$. We note that, for simplicity, we only consider the case of full connectivity. However, as shown in our previous work [44], the mean-field theory also works well for random connectivity with fixed in-degree.

We remark that the model Eq (16a) corresponds to LNP neurons with an exponential linear filter, which we have chosen in the Results part for simplicity. The mesoscopic theory developed here can readily be extended to LNP neurons described by a general linear filter $\kappa(t)$. In this case and assuming $\hat{h}_i(0) = 0$, the $h$-dynamics will be given by

$$\hat{h}_i(t) = \int_0^{t^+} \kappa(t - t')\left(\frac{\mu(t')}{\tau} + \frac{J}{N}\sum_{j=1}^{N} \hat{u}_j(t'^-)\hat{x}_j(t'^-)s_j(t')\right) dt'. \tag{17}$$

With this extension more realistic neuronal dynamics can be modeled [54]. The simple

dynamics Eq (16a) is recovered if the linear filter is chosen as $\kappa(t) = e^{-t/\tau}\theta(t)$, where $\theta(t)$ is the Heaviside step function.

**Mesoscopic model.** In the Appendix B of [44], it has been shown that the mesoscopic dynamics of the empirical variables

$$h(t) := \frac{1}{N}\sum_{i=1}^{N}\hat{h}_i(t), \quad u(t) := \frac{1}{N}\sum_{i=1}^{N}\hat{u}_i(t), \quad x(t) := \frac{1}{N}\sum_{i=1}^{N}\hat{x}_i(t),$$

$$P(t) := \frac{1}{N}\sum_{i=1}^{N}\hat{u}_i^2(t), \quad Q(t) := \frac{1}{N}\sum_{i=1}^{N}\hat{x}_i^2(t), \quad R(t) := \frac{1}{N}\sum_{i=1}^{N}\hat{u}_i(t)\hat{x}_i(t) \tag{18}$$

can be approximated in discrete time with a macroscopic infinitesimal time step $\Delta t$ (as defined above) by the moment-closure equations

$$h_{k+1} = h_k + \frac{\mu_k - h_k}{\tau}\Delta t + \frac{J}{N}\left[R_k\Delta n_k + (u_k\varepsilon_k^x + x_k\varepsilon_k^u)\sqrt{\Delta n_k}\right], \tag{19a}$$

$$u_{k+1} = u_k + \frac{U_0 - u_k}{\tau_F}\Delta t + \frac{U}{N}\left[(1 - u_k)\Delta n_k - \varepsilon_k^u\sqrt{\Delta n_k}\right] \tag{19b}$$

$$x_{k+1} = x_k + \frac{1 - x_k}{\tau_D}\Delta t - \frac{1}{N}\left[R_k\Delta n_k + (u_k\varepsilon_k^x + x_k\varepsilon_k^u)\sqrt{\Delta n_k}\right], \tag{19c}$$

$$P_{k+1} = P_k + 2\frac{U_0 u_k - P_k}{\tau_F}\Delta t + \frac{1}{N}\left[\mu^P(u_k)\Delta n_k + \varepsilon_k^P\sqrt{\Delta n_k}\right], \tag{19d}$$

$$Q_{k+1} = Q_k + 2\frac{x_k - Q_k}{\tau_D}\Delta t + \frac{1}{N}\left[\mu^Q(u_k, x_k, P_k, Q_k, R_k)\Delta n_k + \varepsilon_k^Q\sqrt{\Delta n_k}\right], \tag{19e}$$

$$R_{k+1} = R_k + \frac{U_0 x_k - R_k}{\tau_F}\Delta t + \frac{u_k - R_k}{\tau_D}\Delta t + \frac{1}{N}\left[\mu^R(u_k, x_k, P_k, R_k)\Delta n_k + \varepsilon_k^R\sqrt{\Delta n_k}\right], \tag{19f}$$

where $u_k = u(k\Delta t)$ and analogous expressions hold for $h_k, x_k, P_k, Q_k, R_k$ and the external input $\mu_k$. In Eq (19), we use the abbreviations

$$\mu^P(u) = U(P(U - 2) - 2u(U - 1) + U),$$

$$\mu^Q(u, x, P, Q, R) = PQ - 2Qu + 2(R + (u - 2)x)(R - ux),$$

$$\mu^R(u, x, P, R) = (U(1 - u)^2 - u^2)x + (U - 1)x(P - u^2) + 2(U(u - 1) - u)(R - ux),$$

and

$$\varepsilon_k^P = 2U(1 + u_k(U - 2) - U)\varepsilon_k^u,$$
$$\varepsilon_k^Q = 2(u_k - 1)x_k^2\varepsilon_k^u + 2u_k(u_k - 2)x_k\varepsilon_k^x,$$
$$\varepsilon_k^R = 2(U(u_k - 1) - u_k)x_k\varepsilon_k^u + (U(1 - u_k)^2 - u_k^2)\varepsilon_k^x.$$

Importantly, the dynamics is driven by two sources of noise: First, $\varepsilon_k^u$ and $\varepsilon_k^x$ are correlated Gaussian random numbers with means $\langle\varepsilon_k^u\rangle = \langle\varepsilon_k^x\rangle = 0$ and (co)variances

$$\langle\varepsilon_k^u\varepsilon_l^u\rangle = (P_k - u_k^2)\delta_{k,l}, \quad \langle\varepsilon_k^x\varepsilon_l^x\rangle = (Q_k - x_k^2)\delta_{k,l}, \quad \langle\varepsilon_k^u\varepsilon_l^x\rangle = (R_k - u_k x_k)\delta_{k,l}, \tag{20}$$

where $\delta_{k,l}$ is the Kronecker delta. The random numbers $\varepsilon_k^u$ and $\varepsilon_k^x$ reflect the heterogeneity of $\hat{u}_i$ and $\hat{x}_i$ across synapses $i = 1, \ldots, N$, respectively. Second, $\Delta n_k$ represents the total spike count in the time step $\Delta t$ which is drawn independently from a Poisson distribution with mean $Nf(h_k)\Delta t$:

$$\Delta n_k \sim \mathrm{Pois}[Nf(h_k)\Delta t]. \tag{21}$$

This equation closes the discrete mesoscopic dynamics with STP derived in [44].

We will now use the discrete dynamics to derive a Langevin equation in continuous time. Because of our assumption that $\Delta t$ is large enough such that it contains many spikes, i.e. $\langle \Delta n_k \rangle \equiv Nf(h_k)\Delta t \gg 1$, we can use again a Gaussian approximation. Thus, we write $\Delta n_k \approx Nf(h_k)\Delta t + \sqrt{Nf(h_k)}\Delta W_k$, where $\Delta W_k$ is an independent, normally distributed random number with $\langle \Delta W_k \rangle = 0$ and $\langle \Delta W_k^2 \rangle = \Delta t$. Furthermore, the noise terms appearing in Eq (19) can be written within a Gaussian approximation as

$$\varepsilon_k^u \sqrt{\Delta n_k} \approx \sqrt{N(P - u^2)f(h_k)}\Delta W_k^u, \qquad \varepsilon_k^x \sqrt{\Delta n_k} \approx \sqrt{N(Q - x^2)f(h_k)}\Delta W_k^x, \tag{22}$$

where we neglected terms of order $\mathcal{O}(N^{\frac{1}{4}})$ and where $\Delta W_k^u$ and $\Delta W_k^x$ are mean-zero Gaussian random numbers with covariance

$$\langle \Delta W_k^u \Delta W_l^u \rangle = \langle \Delta W_k^x \Delta W_l^x \rangle = \delta_{k,l}\Delta t, \qquad \langle \Delta W_k^u \Delta W_l^x \rangle = \delta_{k,l}\rho_k \Delta t \tag{23}$$

Here, we introduced the correlation coefficient

$$\rho_k = \frac{R_k - u_k x_k}{\sqrt{(P_k - u_k^2)(Q_k - x_k^2)}}. \tag{24}$$

It follows that, within the Gaussian approximation, the other noise terms are given by

$$\begin{aligned}
\varepsilon_k^P \sqrt{\Delta n_k} &\approx 2U(1 + u_k(U - 2) - U)\sqrt{N(P_k - u_k^2)f(h_k)}\Delta W_k^u, \\
\varepsilon_k^Q \sqrt{\Delta n_k} &\approx 2(u_k - 1)x_k^2 \sqrt{N(P_k - u_k^2)f(h_k)}\Delta W_k^u \\
&\quad + 2u_k(u_k - 2)x_k \sqrt{N(Q_k - x_k^2)f(h_k)}\Delta W_k^x, \\
\varepsilon_k^R \sqrt{\Delta n_k} &\approx 2(U(u_k - 1) - u_k)x_k \sqrt{N(P_k - u_k^2)f(h_k)}\Delta W_k^u \\
&\quad + (U(1 - u_k)^2 - u_k^2)\sqrt{N(Q_k - x_k^2)f(h_k)}\Delta W_k^x.
\end{aligned}$$

Taking the continuum limit $\Delta t \to 0$ yields the Itô stochastic differential equation

$$dh_t = \frac{\mu_t - h_t}{\tau}dt + \frac{J}{N}\left[R_t dn_t + u_t\sqrt{N(Q_t - x_t^2)f(h_t)}dW_t^x + x_t\sqrt{N(P_t - u_t^2)f(h_t)}dW_t^u\right], \quad (25a)$$

$$du_t = \frac{U_0 - u_t}{\tau_F}dt + \frac{U}{N}\left[(1 - u_t)dn_t - \sqrt{N(P_t - u_t^2)f(h_t)}dW_t^u\right] \quad (25b)$$

$$dx_t = \frac{1 - x_t}{\tau_D}dt - \frac{1}{N}\left[R_t dn_t + u_t\sqrt{N(Q_t - x_t^2)f(h_t)}dW_t^x + x_t\sqrt{N(P_t - u_t^2)f(h_t)}dW_t^u\right], \quad (25c)$$

$$dP_t = 2\frac{U_0 u_t - P_t}{\tau_F}dt + \frac{1}{N}\left[\mu^P(u_t)dn_t + 2U(1 + u_t(U - 2) - U)\sqrt{N(P_t - u_t^2)f(h_t)}dW_t^u\right], \quad (25d)$$

$$dQ_t = 2\frac{x_t - Q_t}{\tau_D}dt + \frac{1}{N}\left[\mu^Q(u_t, x_t, P_t, Q_t, R_t)dn_t \right.$$
$$\left. + 2(u_t - 1)x_t^2\sqrt{N(P_t - u_t^2)f(h_t)}dW_t^u + 2u_t(u_t - 2)x_t\sqrt{N(Q_t - x_t^2)f(h_t)}dW_t^x\right], \quad (25e)$$

$$dR_t = \frac{U_0 x_t - R_t}{\tau_F}dt + \frac{u_t - R_t}{\tau_D}dt + \frac{1}{N}\left[\mu^R(u_t, x_t, P_t, R_t)dn_t \right.$$
$$+ 2(U(u_t - 1) - u_t)x_t\sqrt{N(P_t - u_t^2)f(h_t)}dW_t^u$$
$$\left. + (U(1 - u_t)^2 - u_t^2)\sqrt{N(Q_t - x_t^2)f(h_t)}dW_t^x\right]. \quad (25f)$$

with Poisson noise

$$dn_t = \pi(dt, [0, Nf(h_{t^-})]), \quad (25g)$$

where $\pi$ is a two-dimensional Poisson random measure with mean $\langle\pi(ds, dt)\rangle = ds\,dt$ (i.e. $n_t$ is a counting process with conditional intensity $Nf(h_{t^-})$ and $dn_t/dt$ is the associated Dirac delta spike train). Furthermore, $W_t^u$ and $W_t^x$ are Wiener processes, where $W_t^u$ and $W_t^x$ have correlated increments

$$\langle dW_t^u dW_s^u\rangle = \langle dW_t^x dW_s^x\rangle = \delta(t - s)dt\,ds, \quad (26a)$$

$$\langle dW_t^u dW_s^x\rangle = \frac{R_t - u_t x_t}{\sqrt{(P_t - u_t^2)(Q_t - x_t^2)}}\delta(t - s)dt\,ds. \quad (26b)$$

Introducing the Gaussian white-noise processes

$$\xi_x(t) = \sqrt{\frac{(Q_t - x_t^2)f(h_t)}{N}}\frac{dW_t^x}{dt}, \quad (27a)$$

$$\xi_u(t) = \sqrt{\frac{(P_t - u_t^2)f(h_t)}{N}}\frac{dW_t^u}{dt}, \quad (27b)$$

the full stochastic dynamics of the mesoscopic neural-mass model can be rewritten in the form

of a Langevin equation

$$\frac{dh}{dt} = \frac{\mu(t) - h}{\tau} + J[RA(t) + u\xi_x(t) + x\xi_u(t)], \tag{28a}$$

$$\frac{du}{dt} = \frac{U_0 - u}{\tau_F} + U(1 - u)A(t) - U\xi_u(t) \tag{28b}$$

$$\frac{dx}{dt} = \frac{1 - x}{\tau_D} - RA(t) - u\xi_x(t) - x\xi_u(t), \tag{28c}$$

$$\frac{dP}{dt} = 2\frac{U_0 u - P}{\tau_F} + \mu^P(u)A(t) + 2U(1 + u(U - 2) - U)\xi_u(t), \tag{28d}$$

$$\frac{dQ}{dt} = 2\frac{x - Q}{\tau_D} + \mu^Q(u, x, P, Q, R)A(t) + 2(u - 1)x^2\xi_u(t) + 2u(u - 2)x\xi_x(t) \tag{28e}$$

$$\frac{dR}{dt} = \frac{U_0 x - R}{\tau_F} + \frac{u - R}{\tau_D} + \mu^R(u, x, P, R)A(t)$$
$$+ 2(U(u - 1) - u)x\xi_u(t) + (U(1 - u)^2 - u^2)\xi_x(t). \tag{28f}$$

with

$$A(t) = \frac{1}{N}\frac{dn_t}{dt}. \tag{28g}$$

The Gaussian white noise processes are given by their covariance functions

$$\langle \xi_u(t)\xi_u(s) \rangle = \frac{(P_t - u_t^2)f(h_t)}{N}\delta(t - s), \tag{28h}$$

$$\langle \xi_x(t)\xi_x(s) \rangle = \frac{(Q_t - x_t^2)f(h_t)}{N}\delta(t - s), \tag{28i}$$

$$\langle \xi_u(t)\xi_x(s) \rangle = (R_t - u_t x_t)\delta(t - s). \tag{28j}$$

Eq (28) constitutes the jump-diffusion model for the full Tsodyks-Markram STP model with depression and facilitation.

In the case of a general linear filter $\kappa$, the derivation of the mesoscopic STP dynamics does not change. The only difference is that the dynamics for $h$ above needs to be changed to corresponding convolution equations. Therefore, for a general linear filter one only needs to replace Eq (28a) by

$$h(t) = \int_0^{t^+} \kappa(t - \hat{t})\left\{\frac{\mu(\hat{t})}{\tau} + J[R(\hat{t}^-)A(\hat{t}) + u(\hat{t}^-)\xi_x(\hat{t}) + x(\hat{t}^-)\xi_u(\hat{t})]\right\}d\hat{t}. \tag{29}$$

## Mesoscopic model with pure synaptic depression

In order to obtain the jump-diffusion model Eq (4) only with short-term synaptic depression but without facilitation as considered in the Results section, we set $u \equiv U_0$, $P \equiv U_0^2$ and $R \equiv U_0 x$. Then, $\mu^Q(u, x, P, Q, R) = -U_0(2 - U_0)Q$ and $\xi_u(t) \equiv 0$, and the dynamics Eq (28) reduce

to

$$\frac{dx}{dt} = \frac{1-x}{\tau_D} - U_0 x A(t) - U_0 \xi_x(t), \tag{30a}$$

$$\frac{dQ}{dt} = 2\frac{x-Q}{\tau_D} - U_0(2-U_0)Qf(h(t)), \tag{30b}$$

with the Gaussian white noise process $\xi_x(t)$ defined as in Eq (28i). In Eq (30b), we have neglected the noise terms and replaced the population activity $A(t)$ by its mean $f(h(t))$ because they enter the dynamics of $x$ only to order $N^{-3/2}$. Finally, introducing the new mesoscopic variable $y = Q - x^2$, we can combine Eqs (30) and (28i) to arrive at the jump-diffusion model Eq (4).

Again, in the case of a general linear filter $\kappa$, the stochastic differential Eq (4a) for $h(t)$ needs to be replaced by the corresponding integral expression

$$h(t) = \int_0^{t^+} \kappa(t-\hat{t})\left\{ \frac{\mu(\hat{t})}{\tau} + JU_0\left[ x(\hat{t}^-)A(\hat{t}) + \sqrt{\frac{yf(h)}{N}}\xi_x(\hat{t}) \right] \right\} d\hat{t}. \tag{31}$$

**Reduction to a pure diffusion process.**   We can further reduce the jump-diffusion model Eq (4) in the large $N \gg 1$ limit by exploiting the Gaussian approximation of the Poisson shot noise Eq (4d) representing the empirical population activity

$$A(t) = \frac{1}{N}\frac{n(t)}{dt} \approx f(h(t)) + \xi_A(t), \tag{32}$$

where the Gaussian white noise process $\xi_A(t)$ has the auto-correlation function

$$\langle \xi_A(t)\xi_A(s) \rangle = \frac{f(h(t))}{N}\delta(t-s). \tag{33}$$

Consequently, we find that

$$U_0 x A(t) + U_0\xi_x(t) = U_0 x f(h) + U_0[x\xi_A(t) + \xi_x(t)].$$

We can simplify the term in brackets by capitalizing on the fact that $x\xi_A$ and $\xi_x$ are independent Gaussian white noises, whose sum is itself a Gaussian random variable with variance

$$\frac{x^2 f(h)}{N} + \frac{(Q-x^2)f(h)}{N} = \frac{Qf(h)}{N}.$$

Finally, we can replace the term $U_0 x A(t) + U_0\xi_x(t)$ in the $h$- and $x$-dynamics of the jump-diffusion model Eq (4) by

$$U_0 x A(t) + U_0\xi_x(t) = U_0 x f(h(t)) + U_0\frac{Qf(h(t))}{N}\xi(t), \tag{34}$$

where $\xi(t)$ is a Gaussian white noise with auto-correlation function $\langle \xi(t)\xi(s) \rangle = \delta(t-s)$, and we retrieve, in an alternative way, the mesoscopic diffusion model Eq (3) with short-term synaptic depression.

As before, for a general linear filter $\kappa$, the stochastic differential Eq (3a) for $h(t)$ needs to be replaced by the corresponding integral expression

$$h(t) = \int_0^{t^+} \kappa(t - \hat{t}) \left\{ \frac{\mu(\hat{t})}{\tau} + JU_0 \left[ x(\hat{t})f(h(\hat{t})) + \sqrt{\frac{Q(\hat{t}^-)f(h(\hat{t}^-))}{N}} \xi(\hat{t}) \right] \right\} d\hat{t}. \tag{35}$$

For completeness, we also provide the diffusion model in the case of multiple populations. For $\alpha = 1, \ldots, M$, the multi-population diffusion model corresponding to Eq (8) reads:

$$\frac{\mathrm{d}h^\alpha}{\mathrm{d}t} = \frac{\mu^\alpha - h^\alpha}{\tau} + \frac{1}{M} \sum_{\beta=1}^M J_{\alpha\beta} U_0 \left[ x^\beta f(h^\beta) + \sqrt{\frac{Q^\beta f(h^\beta)}{N}} \xi^\beta(t) \right], \tag{36a}$$

$$\frac{\mathrm{d}x^\alpha}{\mathrm{d}t} = \frac{1 - x^\alpha}{\tau_D} - U_0 \left[ x^\alpha f(h^\alpha) + \sqrt{\frac{Q^\alpha f(h^\alpha)}{N}} \xi^\alpha(t) \right], \tag{36b}$$

$$\frac{\mathrm{d}Q^\alpha}{\mathrm{d}t} = 2\frac{x^\alpha - Q^\alpha}{\tau_D} - U_0(2 - U_0)Q^\alpha f(h^\alpha), \tag{36c}$$

with Gaussian white noises $\xi^\alpha(t)$ obeying $\langle \xi^\alpha(t) \rangle = 0$ and $\langle \xi^\alpha(t)\xi^\beta(s) \rangle = \delta_{\alpha,\beta}\, \delta(t - s)$ for all $\alpha, \beta \in \{1, \ldots, M\}$.

## Recurrent network parameters and numerical simulations

In the Results section, we presented and analyzed the network dynamics of a single excitatory population consisting of $N$ LNP-STD spiking neurons following the microscopic dynamics Eq (1) or the mesoscopic dynamics Eqs (3)/(4). For the ring-attractor network model to study hippocampal replay dynamics, we used the microscopic dynamics Eq (7) and the mesoscopic dynamics Eq (8). The parameters to obtain Figs 2–6 are detailed in Table 1. The number $N$ of neurons per population $\alpha = 1, \ldots, M$ is indicated inside the Figures. The model specification and parameters of the Romani-Tsodyks (RT) model for fatigue-induced hippocampal replay, with which we compared our results for the mesoscopic ring-attractor network model in the single and in the multiple environment case in Figs 4–6, respectively, can be found in [21].

**Table 1. Parameters used in Figs 2–6.**

| Parameters | | Fig | 2 | 3 | 4/5 | 6 |
|---|---|---|---|---|---|---|
| Synaptic time constant | $\tau$ | [s] | 0.05 | 0.05 | 0.01 | 0.01 |
| Depression time constant | $\tau_D$ | [s] | 0.8 | 0.6 | 0.8 | 0.8 |
| Utilization of synaptic resources | $U_0$ | | 0.4 | 0.4 | 0.8 | 0.8 |
| Static nonlinearity $f(h)$ of LNP model | | | | | | |
| —Suprathreshold slope | $r$ | | 3.15 | 3.15 | 1.0 | 1.0 |
| —Smoothness at threshold | $\alpha$ | | 0.25 | 0.2 | 1.0 | 1.0 |
| —Threshold | $h_0$ | [mV] | 2.0 | 2.0 | 0.0 | 0.0 |
| Coupling constant | $J \cdot \tau$ | [mV] | 3.5 | 3.5 | – | – |
| Uniform feedback inhibition | $J_0 \cdot \tau$ | [mV] | – | – | 13 | 16 |
| Map-specific interaction | $J_1 \cdot \tau$ | [mV] | – | – | 30 | 25 |
| External input (meso-,microscopic models) | $\mu$ | [mV] | 1.4 | 1.4 | -1.4 | -1.5 |
| External input (only macroscopic model) | $\mu_{\mathrm{macro}}$ | [mV] | – | – | -0.9 | -0.35 |
| Number of populations | $M$ | | 1 | 1 | 100 | 300 |

**Table 2. Simulation results complementing Figs 4 and 5.**

|  | Mesoscopic | Microscopic | RT model | Macro$_{\mu = -0.9}$ |
|---|---|---|---|---|
| $T_{\mathrm{sim}}$ [s] | 4000 | 4000 | 2350 | 2350 |
| # bursts | 5030 | 5040 | 5096 | 5167 |
| slope(# peaks/duration) | 9.28 | 9.26 | 7.80 | 6.85 |
| slope(distance/duration) | 17.27 | 17.17 | 17.56 | 16.81 |
| mean(IBI) | 0.652 | 0.651 | 0.293 | 0.316 |
| CV(IBI) | 0.846 | 0.842 | 0.794 | 0.847 |
| skewness $\gamma_s$(IBI) | 0.917 | 0.940 | 1.230 | 1.224 |
| resc. skewness $\alpha_s$(IBI) | 0.361 | 0.372 | 0.516 | 0.481 |
| kurtosis $\gamma_e$(IBI) | -0.115 | -0.047 | 0.499 | 0.454 |
| resc. kurtosis $\alpha_e$(IBI) | -0.011 | -0.004 | 0.053 | 0.042 |
| # NLE ($>$1 peak) | 1019 | 967 | 939 | 1140 |
| fraction(NLE/bursts) | 20.3% | 19.2% | 18.4% | 22.1% |
| fraction(forward/NLE) | 51.03% | 47.88% | 51.54% | 49.39% |
| mean(abs(NLE speed)) | 12.41 | 12.54 | 12.79 | 12.64 |
| Serial correlations |  |  |  |  |
| Lag 1 (event speed) | 0.048 | 0.054 | -0.563 | -0.344 |
| Lag 1 (forward/backward) | 0.062 | 0.062 | -0.525 | -0.331 |
| Lag 2 (event speed) | -0.043 | -0.010 | 0.434 | 0.229 |
| Lag 2 (forward/backward) | -0.041 | -0.001 | 0.401 | 0.226 |
| Lag 3 (event speed) | -0.013 | 0.053 | -0.320 | -0.074 |
| Lag 3 (forward/backward) | -0.018 | 0.056 | -0.312 | -0.082 |
| Lag 4 (event speed) | -0.010 | 0.011 | 0.227 | 0.021 |
| Lag 4 (forward/backward) | -0.015 | 0.007 | 0.215 | 0.024 |
| Lag 5 (event speed) | 0.008 | -0.032 | -0.139 | 0.058 |
| Lag 5 (forward/backward) | 0.007 | -0.033 | -0.135 | 0.052 |

We computed the statistics for the sequence of interburst intervals (IBI) $T_i$, $i = 1, 2, \ldots$, as follows: the mean is the first cumulant $\kappa_1 = \langle T_i \rangle$; the coefficient of variation (CV) is $\mathrm{CV} = \sqrt{\kappa_2}/\kappa_1$ with second cumulant $\kappa_2 = \langle T_i^2 \rangle - \kappa_1^2$; the skewness is $\gamma_s = \kappa_3 \kappa_2^{-3/2}$ with third cumulant $\kappa_3 = \langle T_i^3 \rangle - 3\kappa_1\kappa_2 - \kappa_1^3$; the rescaled skewness is $\alpha_s = \gamma_s/(3\mathrm{CV}) = \kappa_1\kappa_2/(3\kappa_2^2)$; the kurtosis is $\gamma_e = \kappa_4 \kappa_2^{-2}$ with fourth cumulant $\kappa_4 = \langle T_i^4 \rangle - 4\kappa_1\kappa_3 - 3\kappa_2^2 - 6\kappa_1^2\kappa_2 - \kappa_1^4$; the rescaled kurtosis is $\alpha_e = \gamma_e/(15\mathrm{CV}^2) = \kappa_1^2\kappa_4/(15\kappa_2^3)$. Similar to the definition of the CV, for which the Poisson process serves as a reference for the IBI variability with CV = 1, the rescaled skewness $\alpha_s$ and rescaled kurtosis $\alpha_e$ use the inverse Gaussian distribution as a reference. Values of $\alpha_s$ larger (smaller) than 1 and $\alpha_e$ larger (smaller) than 0 indicate that the IBI distribution is more (less) skewed and more (less) peaked, respectively, than an inverse Gaussian, see [171].

We performed numerical simulations of the microscopic, mesoscopic and macroscopic dynamics using an Euler–Maruyama scheme with time step $dt = 10^{-4} s$. In the single-population scenario considered in Figs 2 and 3, we ran the simulations for $T_{\mathrm{sim}} = 100'000 s$ to obtain significant statistics.

**A circular environment.** In the ring-attractor network with a single environment stored in the synaptic connectivity, we ran the simulations long enough to have at least 5'000 burst events, which allows for a meaningful comparison of the different models. In Table 2, we list the simulation length $T_{\mathrm{sim}}$ together with an overview over the simulation results.

In more detail, we defined burst events as contiguous epochs of the averaged population activity above a certain threshold (taken as the activity averaged across neurons and the whole simulation period). There is an almost perfect agreement between the microscopic network and our mesoscopic description: First, the number of bursts (5'040 vs. 5'030) coincides up to an error of less than 0.2%. Second, the slopes of the linear regression between the number of peaks and the duration per burst are almost the same, see also Fig 4Dii and 4Eii, and our

model prediction ($\sim 9.3$) is closer to the value observed experimentally ($\sim 10.0$, [83, 92]) than the macroscopic model predictions. Third, linear regression between the distance traveled during an event and its duration, yield almost identical slopes for the micro- and the mesoscopic models (Fig 4Diii and 4Eiii). Fourth, also the means and the coefficients of variation (CV) of the interburst intervals, i.e. the time from the end of the $k$th burst until the start of the $k + 1$st burst, coincide. Next, we defined nonlocal replay events (NLEs) as bursts of the average activity with more than one peak so that a transient traveling wave is visible in the density plots in Fig 4Bii and 4Cii, resembling a hippocampal replay pattern. Among all bursts events, around 20% are NLEs, which is consistent across all four models. Around half of all the NLEs travel in anti-clockwise/negative direction ("forward replay") and the other half in clockwise/positive direction ("backward replay" or "preplay"); also this feature is consistent across the different models and matches experimental observations, where in individual experimental sessions both forward and backward replay events are detected with similar proportion [83, 92]. To determine the absolute event speed per NLE, see Fig 4Div and 4Eiv, we divided the distance of the traveled path (irrespective of forward or backward direction) by the duration of the NLE. The means of the absolute NLE speeds in all four models were again close to each other, which stresses the robustness of our findings across parameters and models. Finally, we also investigated serial correlations of the NLEs. Positive serial correlations of the event speed (now taking into account also the travel direction by considering negative speed for backward replays) at lag $n$ indicate that the $k$th and the $k + n$th NLE are more likely to travel in the same direction, whereas negative correlations indicate these NLEs to travel in opposite directions. While the macroscopic and the RT model showed strong negative correlations at lag 1 and positive correlations at lag 2, correlations in the mesoscopic and microscopic models were negligible, see also Fig 5B. Besides, computing the serial correlations not for the (directional) event speed, but for a binary vector of forward/backward replay directions (by taking the sign of the directional event speed) yielded comparable results (see Table 2).

**Multiple circular environments.**   In the ring-attractor network with three circular environments stored in the synaptic connectivity, we ran micro- and mesoscopic simulations long enough to match the number of bursts in the deterministic (macroscopic and RT) model simulations of length $T_{\text{sim}} = 1'000s$, see Table 3. To reduce confounding asymmetries in the underlying synaptic connectivity structure, we constructed the selectivity vectors $\zeta_\alpha$ in Eq (10) pseudo-randomly while guaranteeing that the number of bursts was equally distributed across the three environments, see Table 3. More precisely, we created the binary selectivity vectors $\zeta_\alpha$ under the constraint that exactly $fM = 90$ of in total $M = 300$ units $\alpha = 1, \ldots, M$ are selective for each environment $k \in \{1, 2, 3\}$. Out of these 90 selective units for environment $k$, units 1, $\ldots$, 7 were selective for all three environments, units 8, $\ldots$, 17 were selective for environments $k$ and $j$ and units 18, $\ldots$, 27 for environments $k$ and $l$ with $j, l \in \{1, 2, 3\}$, $k \neq j \neq l \neq k$. The remaining 63 units were exclusively selective for environment $k$. After this selection process, we randomly shuffled these 90 units and drew unique, evenly distributed place field angles $\theta_\alpha^k \in \{2\pi/90, 4\pi/90, \ldots, 2\pi\}$ for each unit $\alpha = 1, \ldots, 90$ selective for environment $k$.

**Table 3. Simulation results complementing Fig 6.**

|  | Mesoscopic | Microscopic | RT model | Macro$_{\mu = -0.35}$ |
|---|---|---|---|---|
| $T_{\text{sim}}$ [s] | 1350 | 1350 | 1000 | 1000 |
| # bursts | 2291 | 2296 | 2354 | 2259 |
| bursts in env. 1 | 34.57% | 33.71% | 32.71% | 33.24% |
| bursts in env. 2 | 31.12% | 31.66% | 34.45% | 33.95% |
| bursts in env. 3 | 34.31% | 34.63% | 32.84% | 32.80% |

To quantify which subsequences occurred more frequently than others, we computed the probabilities for subsequences with three distinct environments $(k, j, l) \in \{1, 2, 3\}^3$ with $k \neq j \neq l \neq k$ by dividing the number of occurrences of a particular subsequence by the number of all possible 3-sequences (= number of all bursts−2). The results are shown in Fig 6.

## Supporting information

**S1 Text. Supplementary figures and tables.** Fig A: shows "Fatigue-induced hippocampal replay in the macroscopic and in the Romani-Tsodyks model" in correspondence to finite-size-induced replay shown in Fig 4. Fig B: shows that "Up-Down dynamics depend on network size" and complements the discussion around Fig 3. Table A: provides a comparison between the simulation results for "Fatigue-induced hippocampal replay dynamics" in the micro-, meso- and macroscopic models. Table B: provides a comparison between the simulation results for "Finite-size-induced hippocampal replay dynamics" in the micro- and mesoscopic models.
(PDF)

## Author Contributions

**Conceptualization:** Bastian Pietras, Valentin Schmutz, Tilo Schwalger.

**Data curation:** Bastian Pietras, Tilo Schwalger.

**Formal analysis:** Bastian Pietras, Valentin Schmutz, Tilo Schwalger.

**Investigation:** Bastian Pietras, Valentin Schmutz, Tilo Schwalger.

**Methodology:** Valentin Schmutz, Tilo Schwalger.

**Software:** Bastian Pietras.

**Supervision:** Tilo Schwalger.

**Visualization:** Bastian Pietras.

**Writing – original draft:** Bastian Pietras, Valentin Schmutz, Tilo Schwalger.

**Writing – review & editing:** Bastian Pietras, Valentin Schmutz, Tilo Schwalger.

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
