## [Decision Letter · Decision Letter 0]

21 Sep 2022

Dear Dr. Schwalger,

Thank you very much for submitting your manuscript "Mesoscopic description of hippocampal replay and metastability in spiking neural networks with short-term plasticity" for consideration at PLOS Computational Biology.

As with all papers reviewed by the journal, your manuscript was reviewed by members of the editorial board and by several independent reviewers. In light of the reviews (below this email), we would like to invite the resubmission of a significantly-revised version that takes into account the reviewers' comments.

All the reviewers agreed that the manuscript contains new and interesting results potentially suitable for publication in this journal. The reviewers' main concern is that the paper is too long and, in particular, the Results section can be significantly shortened. Moreover, parts of the Results are hard to understand and exceedingly technical in jargon. We thus recommend that the authors revise the text following the reviewers' suggestions, clarifying the model, and paying special attention in making the presentation more accessible for a general audience. A few technical clarifications are also necessary, as suggested by the reviewers.

We cannot make any decision about publication until we have seen the revised manuscript and your response to the reviewers' comments. Your revised manuscript is also likely to be sent to reviewers for further evaluation.

Sincerely,

Luca Mazzucato

Guest Editor

PLOS Computational Biology

Lyle Graham

Section Editor

PLOS Computational Biology

All the reviewers agreed that the manuscript contains new and interesting results potentially suitable for publication in this journal. The reviewers' main concern is that the paper is too long and, in particular, the Results section can be significantly shortened. Moreover, parts of the Results are hard to understand and exceedingly technical in jargon. We thus recommend that the authors revise the text following the reviewers' suggestions, clarifying the model, and paying special attention in making the presentation more accessible for a general audience. A few technical clarifications are also necessary, as suggested by the reviewers.

Reviewer's Responses to Questions

**Comments to the Authors:**

Reviewer #1: The authors introduce a theoretical framework that allows one to derive the dynamics of homogeneous but finite-size neuronal populations, in presence of short-term synaptic depression. The “finite-sizeness” (if I can say so) turns out to be important, because it results in the statistics of the “noise” at the mesoscopic level, which is rigorously derived from the microscopic description. This is to be contrasted with standard approaches where the noise (typically white) is added on a phenomenological basis. The framework is then applied to two different models (a model of up-down dynamics and a model of hippocampal-replay dynamics) and it is shown to be very accurate from a quantitative point of view. In both cases, the impact of finite-size noise is carefully studied.

The manuscript contains interesting results, mostly from a theoretical/mathematical point of view, but the numerical simulations presented could (should) be of interest also for more experimentally-oriented readers. The results are novel as far as I can say. The manuscript also includes a lucid discussion of the limitations of the presented results. Overall, it is clearly written though, perhaps, a little too long.

I recommend publication.

Reviewer #2: My detailed review has been uploaded as a PDF attachment

Reviewer #3: The manuscript presents a derivation of a mesoscopic mean-field dynamics for a network of LNP-STD (linear-nonlinear Poisson with short-term plasticity/depression) spiking neurons capable to describe the evolution of the mean membrane potential and the mean fraction of available synaptic resources as in (Holcman and Tsodyks, 2006 – cited [HT06]). Resorting a temporal coarse-graining a la Gillespie, the authors incorporate a state- and size-dependent driving noise whose fluctuations are modulated by the size N, the second moment Q and the gain function f (the activity level). They further extend this mesoscopic description to the case in which the number of neurons is relatively small such that the assumptions underlying the diffusion approximation are broken, eventually deriving a hybrid-noise version of the mesoscopic dynamics. As application, the authors study hippocampal replay in a model network composed of such stochastic mesoscopic populations. The aim is to prove that differently to previous fatigue-driven deterministic models (Romani and Tsodyks, 2015 – cited [RT15]), on longer timescales they differ from the one introduced here as relying on fluctuation-driven metastability.

Overall I found this work interesting and sound, and surely it deserves to be published in PLoS Computational Biology. My feeling is that some parts of the paper appear to be not necessary, as detailed below together with a list of concerns to be addressed before the manuscript is published:

1. Why is the hybrid-noise version of the mesoscopic dynamics needed?

At page 12 the authors write “In the following, we use the simpler diffusion model because it already faithfully reproduces microscopic simulations.” So for a large part of this paper the mesoscopic dynamics (4) does not seem to be useful to the aims of the authors.

Furthermore, I found some degree of contradiction in using the purely diffusive dynamics (3) for the mesoscopic model describing the hippocampal replay, where the size of the single populations is N = 50. This is exactly the range of neuron number presented in Fig. 2 (N = 20 vs N = 200) and Fig. 3 (N = 100) where the hybrid-noise version works better. One could interpret the chosen presentation of the results as a proof of the fact that, when at the multiscale level, the hybrid-noise system does not provide any quantitative advantage. If this is the case, why to present it in this paper?

2. Which is the advantage of having a state-dependent finite-size noise?

In [HT06] the mean-field dynamics is stochastic but is not state-dependent. Here the authors are able to derive from the microscopic dynamics the same equation but with a state-dependent noise. It is indeed modulated not only by the size N but also by the moment Q multiplied by f. Is this an added value by itself? I mean, having a state-dependent noise leads to have qualitatively different collective dynamics compared to what presented in [HT06]? To this purpose it would be interesting to see the time course of Q(t)*f(h) in Figure 2 or 3. A direct comparison with the case of a constant size of the noise (i.e., the \\sigma of [HT06]) would be useful. The result of this investigation could contribute to make more specific the comparison with the model introduced in [HT06] whose related text at page 9-10 should be moved to Discussion section, I would suggest.

3. Again on the paper length.

I often found the Results subsections having part of the text more suited for the Introduction (see the beginning of “Mesoscopic model for hippocampal replays” page 11) or for the Discussion (see the second half of the subsection “Up-Down dynamics in an excitatory population” pages 10-11).

The authors might consider the possibility to make the text leaner and less redundant by moving part of the Results in the Introduction and Discussion sections. About short-term facilitation: is this part of the text really needed?

4. Repetition of waves along the circle.

At page 13 the authors write “a single wave of activity never travels further than once around the circle”. This point should be further investigated as it is related to the critical interplay between spatial and temporal scales in the model. Indeed, I guess this happens because the speed of propagation of the wave is so fast that the duration of a NLE is of the same order of the decay time \\tau_D of the synaptic resources. What happens if the wave speed is lower, less populations are included in the model (smaller M) or a faster recovery from STD (smaller \\tau_D) is taken into account? What about the stability of the Up state weakened by decreasing the Js? This is another degree of freedom affecting the spatial and temporal extension of the NLE.

Furthermore, the authors should consider the fact that noise-driven onset of travelling waves has been studied in model and experiment also in (Capone et al., Cereb. Cortex 2019 – not cited), a similar mechanism which I would suggest to mention at least in the Discussion.

5. Advantage of the mesoscopic modeling compared to the microscopic one.

In Fig. 4 the equivalence between the two descriptions is further proven but I do not see the advantage of using the former compared to the latter. Computational efficiency? From a theoretical point of view, the phase-plane study is performed on the deterministic version of the model, which from Fig. 5 and Fig. S2 seem to be qualitatively similar to the Romani-Tsodyks model. In other words, what one can do with the mesoscopic dynamics that cannot be done directly resorting to the microscopic model?

Minor Points:

1. Page 4: Why LNP-STP? It should be LNP-STD (short-term depression).

2. Page 5: “The trajectories hi(t) are càdlàg …”, is it a misprint?

3. Page 5: “It can also be noted that h_i(t) = h_j(t) [… if the] h_i share the same initial condition.” I guess this is not correct, as h_i dynamics is determined by a stochastic input related to s_i which is different from neuron to neuron: Am I right?

If this is the case I would suggest to change the title of the subsection “Diffusion approximation for the mesoscopic dynamics with short-term depression” in the Methods, considering the possibility to write “Mean-field approximation for …”.

4. Page 23: The statement “conditioned on h(t^-), all the variables … are independent” should be clarified. Here there is the mean-field approximation at work, as from Eq. (1c) the spike train of the i-th neuron depend on f(h_i) and not on f(h).

5. Page 24: “To close the system Eq. (12)”, remove “Eq.”.

6. Write in the figure captions (Fig. 2-6) that the model parameters can be found in Table 1.

7. Page 8: If I am not wrong, the bistable network in the limit N -> \\infty is the same as in [HT06]. I would suggest to write it explicitly at the beginning of the “Up-Down dynamics in an excitatory population”.

8. The second part of the subsection “Up-Down dynamics in an excitatory population” appears to be more for the Discussion, I would suggest to compress it.

9. Fig. 4 caption: what are the red lines in Panels D and E? Linear regression curves?

10. Fig. 5 and 6: Unit measures in the color bars of panel 5Ci and 6D are not shown: Are they [Hz]?

11. Page 15: “Assuming that each peak in the average activity corresponds to a SWR”. SWR are oscillatory events at specific high frequencies and if I am not wrong, they are not explicitly observed in the model presented in this paper. This statement should than be made more clear and detailed, otherwise I would suggest to remove this reference to SWRs.

12. Page 21/22: About the conductance-based models, citations may appear excessively biased. For instance, the sentence “For an extension of the mesoscopic model to a conductance-based description of synaptic input, the interested reader is referred to the discussion of [26].” The authors may consider also to cite the work from the Destexhe’s lab (di Volo et al., Neural Comput 2019 – not cited) and (Capone et al., Phys Rev E 2019 – not cited).

**Have the authors made all data and (if applicable) computational code underlying the findings in their manuscript fully available?**

Reviewer #1: Yes

Reviewer #2: Yes

Reviewer #3: Yes

PLOS authors have the option to publish the peer review history of their article (what does this mean?). If published, this will include your full peer review and any attached files.

Reviewer #1: No

Reviewer #2: No

Reviewer #3: **Yes: **Maurizio Mattia
---

## [Editor Report · Decision Letter 1]

11 Dec 2022

Dear Dr. Schwalger,

We are pleased to inform you that your manuscript 'Mesoscopic description of hippocampal replay and metastability in spiking neural networks with short-term plasticity' has been provisionally accepted for publication in PLOS Computational Biology.

Best regards,

Luca Mazzucato

Guest Editor

PLOS Computational Biology

Lyle Graham

Section Editor

PLOS Computational Biology

---

## [Editor Report · Acceptance letter]

16 Dec 2022

PCOMPBIOL-D-22-00667R1 

Mesoscopic description of hippocampal replay and metastability in spiking neural networks with short-term plasticity

Dear Dr Schmutz,

I am pleased to inform you that your manuscript has been formally accepted for publication in PLOS Computational Biology. Your manuscript is now with our production department and you will be notified of the publication date in due course.

With kind regards,

Anita Estes
